# Dated gene duplications elucidate the evolutionary assembly of eukaryotes

Christopher J. Kay[1,2 ✉], Anja Spang[3,4], Gergely J. Szöllősi[5,6,7], Davide Pisani[2 ✉], Tom A. Williams[2,8 ✉] & Philip C. J. Donoghue[1 ✉]

The origin of eukaryotes was a formative but poorly understood event in the history of life. Current hypotheses of eukaryogenesis differ principally in the timing of mitochondrial endosymbiosis relative to the acquisition of other eukaryote novelties[1]. Discriminating among these hypotheses has been challenging, because there are no living lineages representative of intermediate steps within eukaryogenesis. However, many eukaryotic cell functions are contingent on genes that emerged from duplication events during eukaryogenesis[2,3]. Consequently, the timescale of these duplications can provide insights into the sequence of steps in the evolutionary assembly of the eukaryotic cell. Here we show, using a relaxed molecular clock[4], that the process of eukaryogenesis spanned the Mesoarchaean to late Palaeoproterozoic eras. Within these constraints, we dated the timing of these gene duplications, revealing that the eukaryotic host cell already had complex cellular features before mitochondrial endosymbiosis, including an elaborated cytoskeleton, membrane trafficking, endomembrane, phagocytotic machinery and a nucleus, all between 3.0 and 2.25 billion years ago, after which mitochondrial endosymbiosis occurred. Our results enable us to reject mitochondrion-early scenarios of eukaryogenesis[5], instead supporting a complexified-archaean, late-mitochondrion sequence for the assembly of eukaryote characteristics. Our inference of a complex archaeal host cell is compatible with hypotheses on the adaptive benefits of syntrophy[6,7] in oceans that would have remained largely anoxic for more than a billion years[8,9].

The origin of eukaryotes was a formative event in the history of life, in which a new kind of cell with distinct functional, morphological and ecological modalities evolved through an evolutionary merger between at least two prokaryotes: an Asgard archaeal host and an alphaproteobacterial endosymbiont[1,2,7,10,11]. How and when eukaryotes originated, and the order in which the eukaryotic characteristics evolved, are the subject of intense debate[2]. Perhaps the most contentious distinction between competing hypotheses of eukaryogenesis concerns the relative timing of mitochondrion acquisition and whether or not it was a fundamental prerequisite to all other steps in the evolution of a eukaryote-grade cell. Other points of difference include the number of endosymbiotic partners involved in eukaryogenesis. Although most hypotheses agree on archaeal and alphaproteobacterial ancestry, the syntrophy hypothesis includes an additional, ∂-proteobacterial partner, which is suggested to have served as the host to an endosymbiotic Asgard archaeon (the future nucleus), to explain the bacterial character of the eukaryotic membrane[7]; in such a scenario the alphaproteobacterial ancestor of the mitochondrion entered the symbiosis at a later stage. The serial endosymbiosis hypothesis[12] invokes multiple, transient endosymbiotic partners preceding mitochondrial endosymbiosis to explain the presence of genes of bacterial origin in the eukaryotic nucleus that are not alphaproteobacterial in origin. Testing among these competing hypotheses is challenging because we lack extant taxa that represent intermediate steps of eukaryogenesis.

Questions about the order in which eukaryotic features emerged could be answered by determining when the genes that underpin key eukaryotic traits were acquired by proto-eukaryotes[2,3,12]. Estimating the age of eukaryote novelties is complicated because gene trees do not have a node corresponding to mitochondrial acquisition; instead, they provide an estimate of the genetic distance to the last common ancestor of the eukaryotic and prokaryotic versions of a gene[1]. Furthermore, eukaryotes have at least two stem lineages: one emerging from within Archaea (nuclear first eukaryotic common ancestor (nFECA)) and the other from within Bacteria (mitochondrial first eukaryotic common ancestor (mFECA))[1]. However, gene duplication is a hallmark of eukaryotes[2,13], and previous work has shown that many genes that underpin important features of eukaryotic biology in the nucleus, cytoskeleton and mitochondrion underwent gene duplications in the eukaryote stem lineages, prior to the last eukaryotic common ancestor (LECA)[2,12]. In many cases, the paralogues arising from these pre-LECA duplications

[1]Bristol Palaeobiology Group, School of Earth Sciences, University of Bristol, Bristol, UK. [2]Bristol Palaeobiology Group, School of Biological Sciences, University of Bristol, Bristol, UK. [3]Department of Marine Microbiology and Biogeochemistry, Royal Netherlands Institute for Sea Research (NIOZ), Den Burg, The Netherlands. [4]Department of Evolutionary and Population Biology, Institute for Biodiversity and Ecosystem Dynamics (IBED), University of Amsterdam, Amsterdam, The Netherlands. [5]MTA-ELTE "Lendulet" Evolutionary Genomics Research Group, Budapest, Hungary. [6]Institute of Evolution, Centre for Ecological Research, Budapest, Hungary. [7]Model-Based Evolutionary Genomics Unit, Okinawa Institute of Science and Technology Graduate University, Okinawa, Japan. [8]Department of Life Sciences, University of Bath, Bath, UK. ✉e-mail: chris.kay@bristol.ac.uk; davide.pisani@bristol.ac.uk; tom.a.williams@bristol.ac.uk; phil.donoghue@bristol.ac.uk

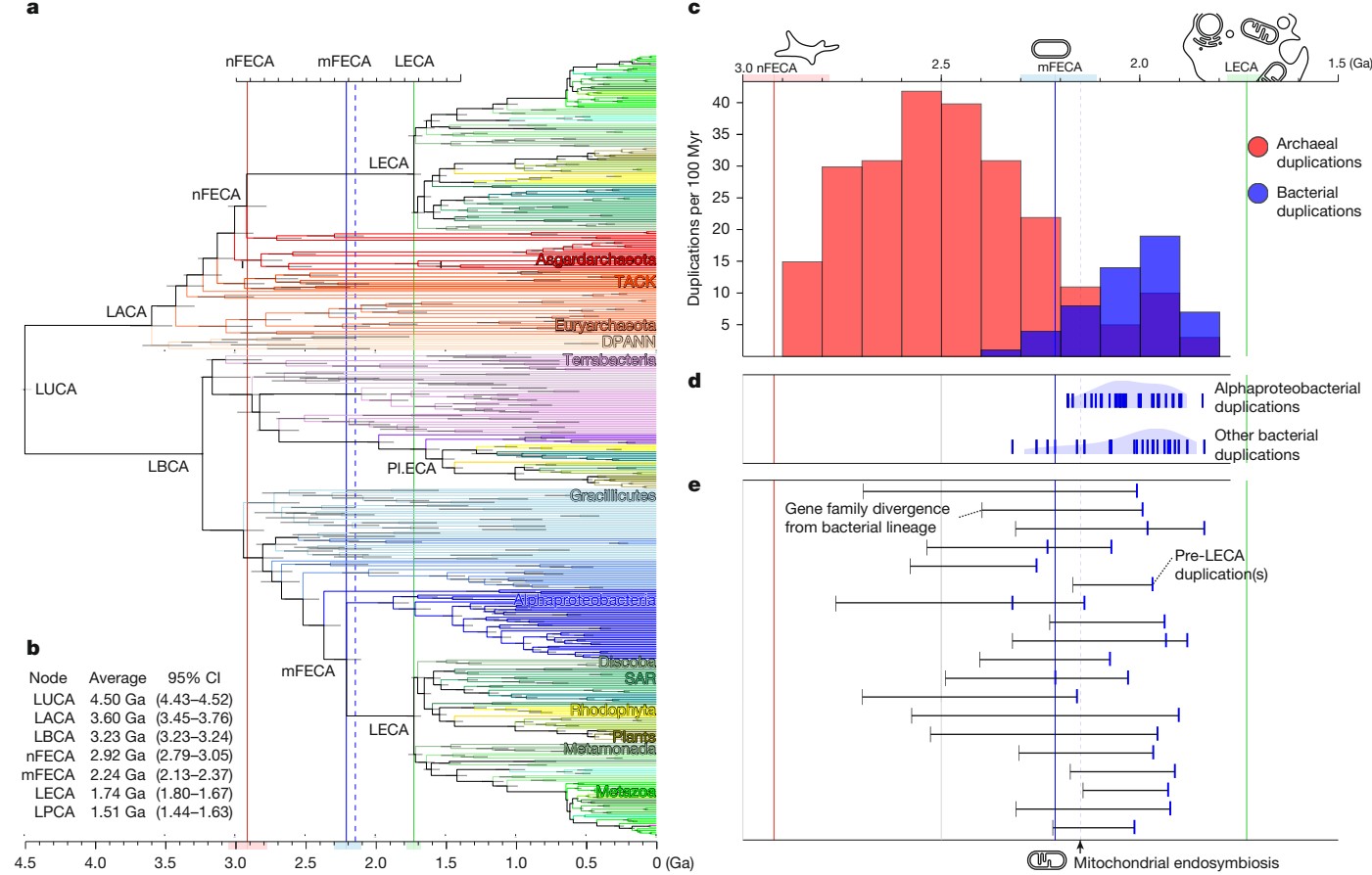

**Fig. 1 | Time-resolved species tree and gene duplications by prokaryotic origin. a**, A time-resolved tree of life. A cross-braced tree was produced using MCMCTree calibrated with 18 fossil calibrations (Supplementary Note 4). Nuclear and mitochondrial eukaryotic clades were asserted to have the same topology and root, and the plastid clade has a distinct topology resulting from secondary plastid acquisition events. Equivalent nodes containing the same species were cross-braced (as in ref. 38)—that is, fixed to the same age. LPCA, last plastid common ancestor; LUCA, last universal common ancestor. **b**, Important node dates and their confidence intervals. **c**, Time-resolved gene duplications of archaeal and bacterial origins were binned into 100-Myr intervals and overlaid. Among the sample of genes that were amenable to analysis, archaeal duplications are more numerous and began earlier, perhaps reflecting elaboration of the

archaeal host genome prior to mitochondrial endosymbiosis. **d**, Comparison of the individual duplication ages of alphaproteobacterial origin and all other bacterial duplications. The onset of alphaproteobacterial duplications happens rapidly after the divergence of mFECA from other alphaproteobacteria, suggesting a short bacterial stem. We align mitochondrial endosymbiosis to the onset of these duplications at around 2.2 Ga. **e**, Gene families of non-alphaproteobacterial origin are shown by the time of their divergence from their bacterial lineage of origin (black vertical lines), and subsequent pre-LECA duplications (blue vertical lines), joined by a horizontal rule. Duplications of bacterial origin have divergence ages up to 2.9 Ga, but most undergo duplication after mitochondrial endosymbiosis.

have distinct roles in the implementation of eukaryote-specific features, such as the Sm–LSm complexes in the spliceosome[14], actin and the actin-related proteins in the cytoskeleton[15]. These duplications are represented in gene trees by a duplication node that unites two descendant LECA clades. Dating this node provides a maximum estimate for the timing of origin of their paralogue-specific functions, identifying evolutionary events along the two stems. Thus, it should be possible to dissect the sequence of innovations in eukaryogenesis on the basis of the timing of duplication of paralogues associated with eukaryote innovations[1].

Our aim here is to establish a relative and absolute timeline for the evolutionary assembly of the eukaryotic cell. We use relaxed molecular clock methodology to date pre-LECA duplications, in order to constrain when each evolutionary novelty emerged along the nuclear and mitochondrial eukaryote stem lineages (Fig. 1a). To overcome difficulties with a lack of temporal resolution from short single-gene alignments, we use a sequential Bayesian approach to calibrate gene trees using age estimates for speciation nodes obtained from a dated species tree of archaea, bacteria and eukaryotes (following refs. 4,16). Finally, we

compare the fit of our estimates for the sequence of emergence of eukaryotic features against the geologic record and existing models of eukaryogenesis.

## Establishing the eukaryogenesis timeline

Our phylogenetic analyses were based on a concatenation of 62 marker genes including taxa chosen to provide a phylogenetically representative sample of eukaryotes, bacteria and archaea, particularly including the closest relatives of eukaryotes and taxa for which geological calibrations are available. Within the phylogenetic tree, which was estimated under maximum likelihood, nFECA branches within Asgard archaea[17] (also known as Promethearchaeota), sister to Heimdallarchaeia, and mFECA is sister to Alphaproteobacteria to the exclusion of Magnetococcales, consistent with recent analyses[18]. Note that nFECA and mFECA refer to the points at which eukaryotes branched from extant, sampled archaea and bacteria, respectively, on the tree of life. Their identities and ages would change if more closely related genomes on either the archaeal or bacterial side of the eukaryotic family tree were

discovered, or our understanding of the relationships of sampled taxa changed. Our relaxed molecular clock analyses using MCMCTree estimated that the divergence of nFECA occurred 3.05–2.79 billion years ago (Ga), the divergence of mFECA from Alphaproteobacteria occurred at 2.37–2.13 Ga, and LECA radiated 1.80–1.67 Ga (Fig. 1a,b and Supplementary Fig. 1). From the perspective of species phylogeny, the hallmark features of eukaryotes that are of archaeal origin must have arisen between nFECA and LECA (3.05–1.80 Ga), and mitochondrial endosymbiosis must have occurred between the divergence of mFECA and LECA (2.37–1.80 Ga). These results indicate that the two eukaryotic stems are of different lengths, with median durations of approximately 1.1 billion years (Gyr) and 0.6 Gyr, respectively. We performed a range of sensitivity analyses to investigate the robustness of this timeline to alternative tree topologies, relative ages of key deeper nodes (such as last bacterial common ancestor (LBCA) and last archaeal common ancestor (LACA)) and the use of particular fossil calibrations (Methods and Supplementary Note 3); the results indicate that the age of the key nodes for the eukaryogenesis timeline (nFECA, mFECA and LECA, and in particular the time spans between them) are robust to the range of conditions tested (Methods and Supplementary Note 3).

To investigate the timing of gene duplications relative to LECA, nFECA and mFECA, we surveyed a broad sample of eukaryotic, archaeal and bacterial genomes for gene families of prokaryotic ancestry with duplications that occurred within either of the two eukaryote stem lineages. In total, we obtained 135 time-resolved gene trees compatible with this requirement, 95 of archaeal origin and 40 of bacterial origin (Fig. 1c). Among the 40 investigated gene families of bacterial descent, 19 did not provide strong evidence of alphaproteobacterial origin. Similarly, 9 of the 95 archaeal gene families did not support an Asgard archaea origin. The age of all of these duplications relative to LECA, nFECA and mFECA constitute the focus of our study. This was achieved by dating paralogue divergences within constraints on the age of LECA and its archaeal- and alphaproteobacterial-derived total groups, established in our earlier dating of these species divergences.

In addition to duplication and functional divergence of bacterial and archaeal genes, eukaryogenesis also involved the duplication of genes novel to eukaryotes that originated prior to LECA. These genes were not integrated into our analyses because, within the gene trees, the duplication node is also the root and, therefore, sensitive to the root prior used in the molecular clock analysis[19]. Approaches to mitigating these effects—for example, by removing the node of interest from the root of the tree[19] or proposing additional maximum calibrations—are not applicable to pre-LECA duplications of novel eukaryote genes.

## Dating mitochondrial endosymbiosis

Debates about the timing of mitochondrial endosymbiosis relative to the evolution of other hallmark features of eukaryotes, such as the nucleus, have proved difficult to resolve because there is no dateable node on the species tree that corresponds to mitochondrial endosymbiosis. However, when investigating pre-LECA duplications of alphaproteobacterial origin, most duplications occur shortly after mFECA (Fig. 1d). Among modern organisms, gene duplication is much more frequent in eukaryotes than in prokaryotes[20], which may reflect a more permissive population genetic environment in eukaryotes allowing the accumulation of duplicate genes that are initially functionally redundant[21]. Among the earliest alphaproteobacterial-origin duplications are gene families that gave rise to paralogues which today function outside the mitochondrion (for example, exclusively nuclear functioning RNA methylase METTL4, 2.24–2.11 Ga; type Y family polymerases, earliest duplication 2.24–2.12 Ga; and mismatch repair proteins, earliest duplication 2.18–2.08 Ga), so we suggest that the initial accumulation of gene duplications in the mitochondrial lineage evidences the onset, or establishment, of mitochondrial integration into the eukaryotic cell. In further support of this, we find alphaproteobacterial-origin genes coding for proteins involved in the functioning of mitochondrial endosymbiosis (preprotein import system TIM14 and TIM44 gene families, 2.24–1.99 Ga and 2.22–2.00 Ga, respectively) with similarly early duplication dates. The onset of gene duplication fixation and the identity of the earliest duplications of alphaproteobacterial origin suggests that symbiosis between an Asgard archaeon and alphaproteobacterium was probably already underway by about 2.2 Ga, providing a minimum age for mitochondrial endosymbiosis. Within these temporal constraints, 85% of the pre-LECA duplicates of archaeal origin were estimated to have been fixed prior to mitochondrial endosymbiosis, indicating that duplication was prevalent on the nFECA branch, and fixation of duplicate genes was not dependent on a change of metabolic circumstance associated with mitochondrial endosymbiosis. Among 19 pre-LECA duplications of bacterial but not alphaproteobacterial origin, 16 duplications have an average node age that postdates our inferred time of mitochondrial endosymbiosis (Fig. 1d,e). Irrespective of whether these genes were already present in the genome of the mitochondrial symbiont owing to previous horizontal gene transfers (HGTs), or whether they represent independent instances of HGT that postdated mitochondrial symbiosis, the timing of their duplications is consistent with a shift to a higher rate of bacterial gene duplicate fixation after the mitochondrial endosymbiosis.

## The development of key eukaryotic traits

Debates about eukaryogenesis are typically framed around a few key eukaryotic apomorphies: the cytoskeleton, endomembrane system, membrane, mitochondrion and nucleus. Although many of the genes that specify these structures are or appear to be specific to eukaryotes, others are clearly of prokaryotic descent. We dated duplications in gene families that are directly associated with key cellular systems, providing insight into the order in which these components were elaborated during eukaryogenesis.

## The cytoskeleton

Eukaryotes have inherited two cytoskeletal filament types from archaea; actin[22] and tubulin[23]. In prokaryotes, homologues of these and associated gene families perform structural roles related to cell division and morphology[24], suggesting that these ancestral functions extend to all domains of life. Cultivated representatives of extant Asgard archaea[15] have protrusions from the cell body, potentially as a means to engage in metabolic exchange with other cells[25], and specific homologues to eukaryotic cytoskeletal proteins have been identified and characterized[15]. These considerations suggest that nFECA inherited a cytoskeleton that may have been involved in cell division, morphology and/or membrane deformation[15]. The actin and tubulin gene families have previously been observed to have duplicated within the Asgard archaea lineage[15,22], and we find that the paralogues inherited by eukaryotes underwent multiple rounds of duplication and divergence (Fig. 2a) along the eukaryogenesis timeline.

Our results indicate that eukaryotic actins underwent duplication beginning around 2.8 Ga. Among the resulting subfamilies, ARP2 and ARP3, which are necessary for the formation of branched actin filaments in modern eukaryotes, arose via duplications at 2.90–2.47 Ga (ARP3) and 2.72–2.06 Ga (ARP2). As branched filament formation is required for phagocytosis, these dates provide a maximum age for the origin of phagocytosis as carried out by extant eukaryotes. Previous phylogenies of actin-related proteins in eukaryotes and Asgard archaea suggest that the diversification of these actin subfamilies might have already occurred during the radiation of Asgard archaea[15,22,26]. If so, the capacity for forming branched actin filaments might be still older, dating back to the Asgard ancestor of eukaryotes.

Oriented tubulin networks have key roles in eukaryotic cell biology, and the gene family began expanding 2.8–2.2 Ga. Our inferred

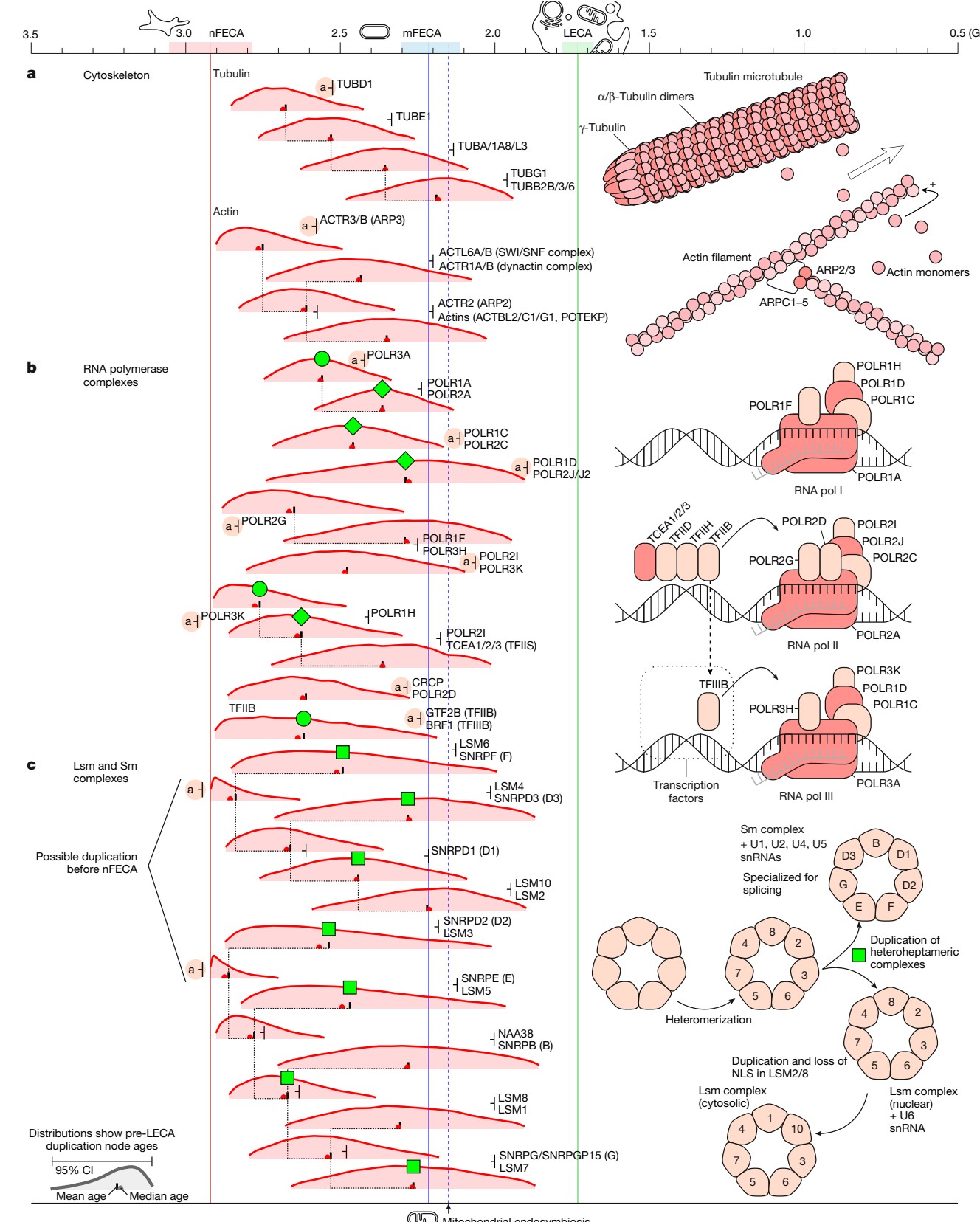

**Fig. 2 | Key gene duplications defining the cytoskeleton and nuclear compartment. a**, Eukaryotes inherited two families of cytoskeletal filament-forming proteins from Asgard archaea, which underwent multiple rounds of duplication before LECA and provide information about the evolution of cytoskeletal complexity. The nuclear compartment is defined by apparently eukaryote-specific gene families, although constraints on nuclear formation can be inferred from duplications in genes whose maturation pathways and function involve transport between compartments. **b,c**, Pre-LECA duplications in the RNA polymerase (**b**) and Sm–LSm (**c**) complexes. Symbols next to each gene family indicate lineage of descent (a, Asgard). Green shapes connect duplication events expected to have a similar age.

early divergence of γ-tubulin (2.84–2.41 Ga, involved with nucleating and orienting microtubules) and δ- and ε-tubulins (2.75–2.25 Ga and 2.61–2.07 Ga, respectively, paralogues localized to centrioles) gives evidence for early establishment of microtubule organization, with the filament-forming α and β paralogues emerging later (2.47–1.92 Ga), suggesting microtubule organization with initially homomeric tubulin filaments. The nuclear spindle, an example of an organized microtubule network, associates with other protein components to segregate chromosomes in cell division. Although ESCRT-III proteins are primarily involved in membrane remodelling and cell division, CHMP4C (duplication date 2.77–2.06 Ga) may additionally be characterized in associating with microtubules[27] and attachment to kinetochores[28], linking microtubule organization to cell division. Together this evidence indicates an organized microtubule network before mitochondrial endosymbiosis, probably by 2.7 Ga.

## The endomembrane system

The endomembrane system is a set of subcellular membrane-bound compartments that carry out distinct functions in the eukaryotic cell. These include the endoplasmic reticulum and Golgi (which together perform membrane protein and lipid biosynthesis and glycosylation) and the endosome, lysosome and autophagosome (collectively, the endolysosomal system which processes extracellular material and recycles other cellular components). Materials are moved between these compartments in vesicles via the cytoskeleton, which our analyses and others[2,25,29,30] suggest developed earlier during eukaryogenesis.

Although duplications of genes involved in the endomembrane vesicle trafficking system occur throughout the timeline of eukaryogenesis (Fig. 3a), the oldest occurred in gene families that traffic between the Golgi, endoplasmic reticulum and plasma membrane, including the SNARE protein STX5 (2.83–2.38 Ga); Rab proteins RAB19, RAB30, RAB33A and RAB33B (2.87–2.26 Ga); RAB6B (2.74–2.07 Ga); YIPF proteins YIPF1 and YIPF2 (2.91–2.55 Ga); the TRAPP protein TRAPPC3 and TRAPPC3L (3.00–2.54 Ga); COPI proteins COPZ1 and COPZ2 (2.87–2.39 Ga); COPD (2.90–2.61 Ga); and COPII proteins SEC23A and SEC23B (2.78–2.33 Ga). Duplications specifying the endosome, lysosome and autophagosome occurred later, beginning around 2.4 Ga and include SNARE proteins STX7 and STX12 (2.63–2.04 Ga); VTI1A and VTI1B (2.75–1.95 Ga); Rab proteins RAB7A, RAB9A and RAB9B (2.74–2.01 Ga); the ARF protein ARF6 (2.55–1.84 Ga); COPI proteins AP3S1 and AP3S2 (2.68–2.11 Ga); and AP3M1 and AP3M2 (2.80–2.31 Ga). From this, we infer that the Golgi, endoplasmic reticulum and plasma membrane were the first endomembrane compartments to be elaborated during eukaryogenesis, whereas the others developed later. The inferred ages suggest that duplication of genes required to pattern new compartments occurred approximately contemporaneously (that is, the 95% HPDs of the duplication ages overlapped; Fig. 3b), consistent with the hypothesis that innovation of new subcellular compartments was achieved by duplication of the genes that specify them[31].

The earliest identified pre-LECA duplications whose gene products are localized exclusively to the endoplasmic reticulum in extant eukaryotes are of archaeal origin (Fig. 4b, divergence of SRD5A1: 2.86–2.25 Ga; DPM2 and PIGP: 2.88–2.06 Ga; STT3A and STT3B: 2.80–2.18 Ga). We were also able to find gene families of bacterial origin that underwent pre-LECA duplication and whose products function in endoplasmic reticulum-localized membrane lipid biosynthesis pathways (ACSL1, GPAT, LPCAT and SPTLC). Some later gene duplications in families of archaeal origin are consistent with integration into these pathways of bacterial origin. For example, SGPP1 and SGPP2 are enzymes of archaeal origin that today carry out an intermediate step in eukaryotic sphingolipid biosynthesis, and we infer their origin via gene duplication diverging from DOLPP1 in the archaeal dolichol glycosylation pathway 2.67–1.98 Ga. Pre-LECA duplications in the other gene families of

bacterial origin that are involved in membrane lipid biogenesis also duplicated at around this time (Fig. 4b, GPAT3, GPAT4 and LPCAT1, LPCAT2 and LPCAT4: 2.54–2.06 Ga; SPTLC1–3: 2.16–1.99 Ga; ACSL1 and ACSL3–6: 2.12–1.87 Ga). Gene trees for two of these families (ACSL and GPAT and LPCAT) do not provide compelling support for an alphaproteobacterial origin, with some (ACSL) consistent with acquisition from other bacterial groups such as Myxococcota[32]. Together, these duplications point to a phase of late complexification in endoplasmic reticulum-associated functions, reflecting metabolic integration of archaeal and bacterial membrane biosynthesis pathways. This finding is in agreement with hypotheses in which the membrane transition involved acquisition of bacterial genes from non-mitochondrial sources.

The endolysosomal system is the digestive part of the endomembrane system, which is responsible for processing endocytosed material and recycling cellular components[10]. The development of the endolysosomal system is particularly relevant to eukaryogenesis because phagocytosis, sometimes invoked as the means by which the archaeal host could have acquired the mitochondrion[33], probably developed from an already established digestive pathway[10]. Several transporter gene families of bacterial descent are found duplicated across the endolysosomal compartments (chloride channel transporters (CLCs), solute carriers (SLCs) and ABCB transporters 1, 2, 5, 9 and 11), where they perform essential roles in compartment function. The oldest duplications in each family (Fig. 4c, CLCN6 and CLCN7: 2.69–2.05 Ga; SLC15A1–5: 2.16–1.82 Ga; ABCB1, ABCB5 and ABCB11: 2.15–1.94 Ga) are coincident with the ages for the development of vesicle trafficking to the endolysosomal compartments, suggesting that they acquired their present day functions contemporaneously with their compartment patterning. Although we identify the ABCB transporters with an alphaproteobacterial origin, the CLC and SLC transporters have separate, apparently non-alphaproteobacterial origins, suggesting that the development of the endolysosomal system required gene sources from bacteria other than alphaproteobacteria.

## The nucleus

Dating the development and closure of the nucleus is key to testing hypotheses of eukaryogenesis (Fig. 2 and Supplementary Figs. 2–4). Many of the key components of the nucleus, such as nuclear pores, appear to be eukaryote-specific innovations that lack homologues in modern prokaryotes. However, several protein complexes that are key to nuclear function developed by duplication from archaeal ancestors, and so the duplication ages of these genes may provide some insight into the development of the nucleus[14]. In extant eukaryotes, the protein components of the spliceosomal Sm and LSm complexes are assembled in the cytosol and imported through nuclear pores into the nucleus. The Sm and LSm complexes evolved from a common homomeric ancestor of archaeal origin[14]. During eukaryogenesis, this ancestral complex underwent multiple rounds of gene duplication, with inferred divergence times ranging 2.91–1.87 Ga (Fig. 2b). A final duplication event gave rise to the distinct Sm and LSm complexes of the modern spliceosome. In our analysis, the nuclear localization signals acquired in two subunits (LSM2 and SMB, and LSM8 and SMD1) occurred prior to complex duplication, which we date to 2.87–1.87 Ga (Fig. 2c, green squares), although we note the possibility that nuclear localization signals might have evolved before the nuclear compartment[34]—for example, to localize these proteins to nucleic acids. Efficient biogenesis of the modern LSm complex requires the nuclear localization of its transient RNA partner U6 snRNA and, more broadly, the spatial separation of transcription and translation with or without a nuclear compartment appears critical for the emergence of regulated splicing. The early timing we infer for the evolution of spliceosomal components therefore supports the emergence of complex eukaryotic splicing before the mitochondrial endosymbiosis.

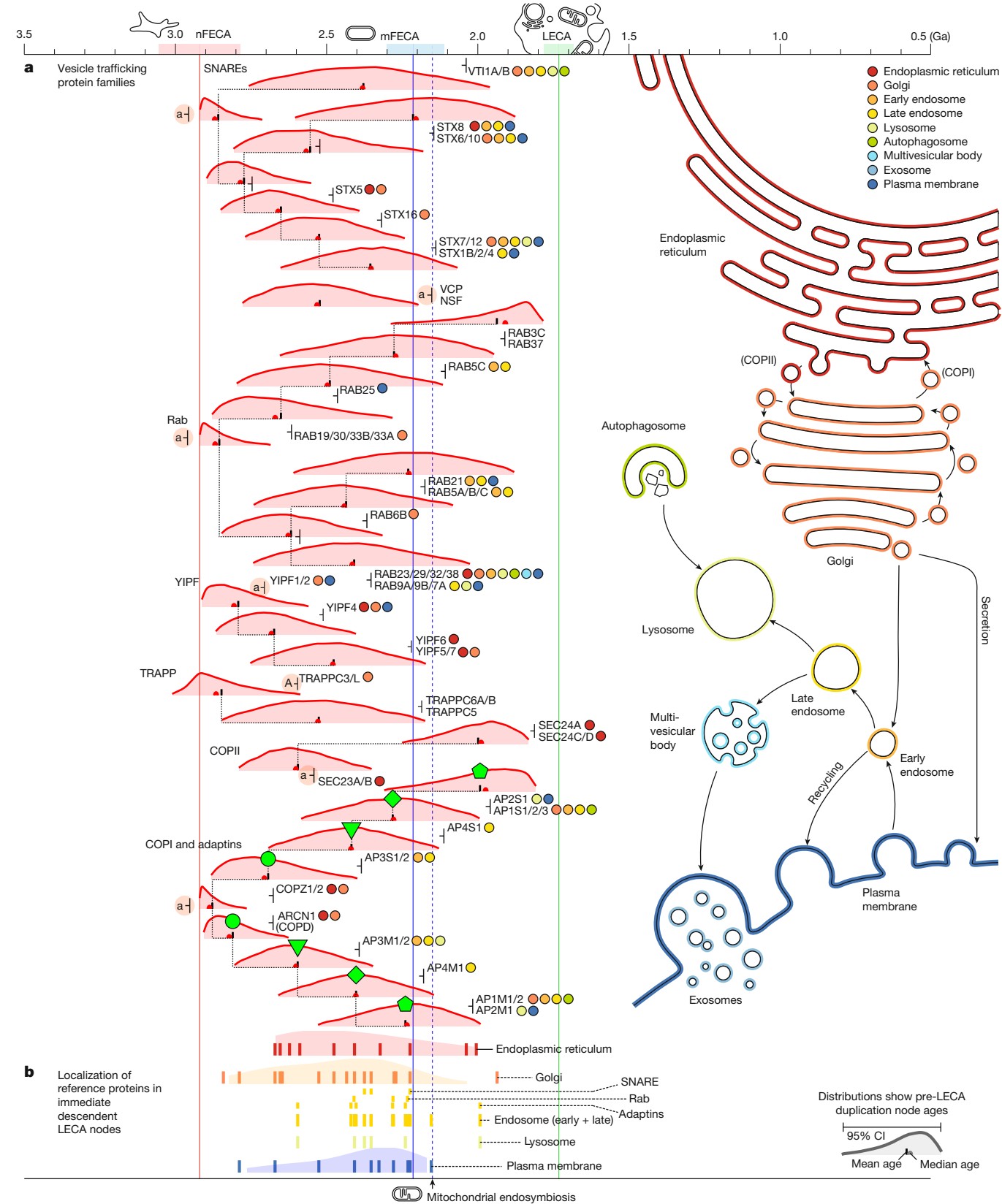

**Fig. 3 | Pre-LECA duplications in eukaryotic vesicle trafficking proteins.**
Several gene families of archaeal ancestry participate in the processes of
material transport between components of the endomembrane system of
extant eukaryotes. **a**, Gene families of archaeal ancestry involved in budding,
fusion and targeting of endomembrane vesicles. By considering the function
of the LECA nodes descending from a duplication (colour coded), we infer that
complexification of the endoplasmic reticulum, Golgi and plasma membrane
preceded the evolution of functions specific to the endolysosomal system.
**b**, Furthermore, we observe that distinct gene families that participate in the
same endolysosomal compartment appear to have duplicated at around the
same time (95% confidence intervals overlap). Format of the age distributions are
as in Fig. 2. Green shapes connect duplication events expected to have a similar
age. Symbols next to each gene family indicate lineage of descent (a, Asgard; A,
other archaea).

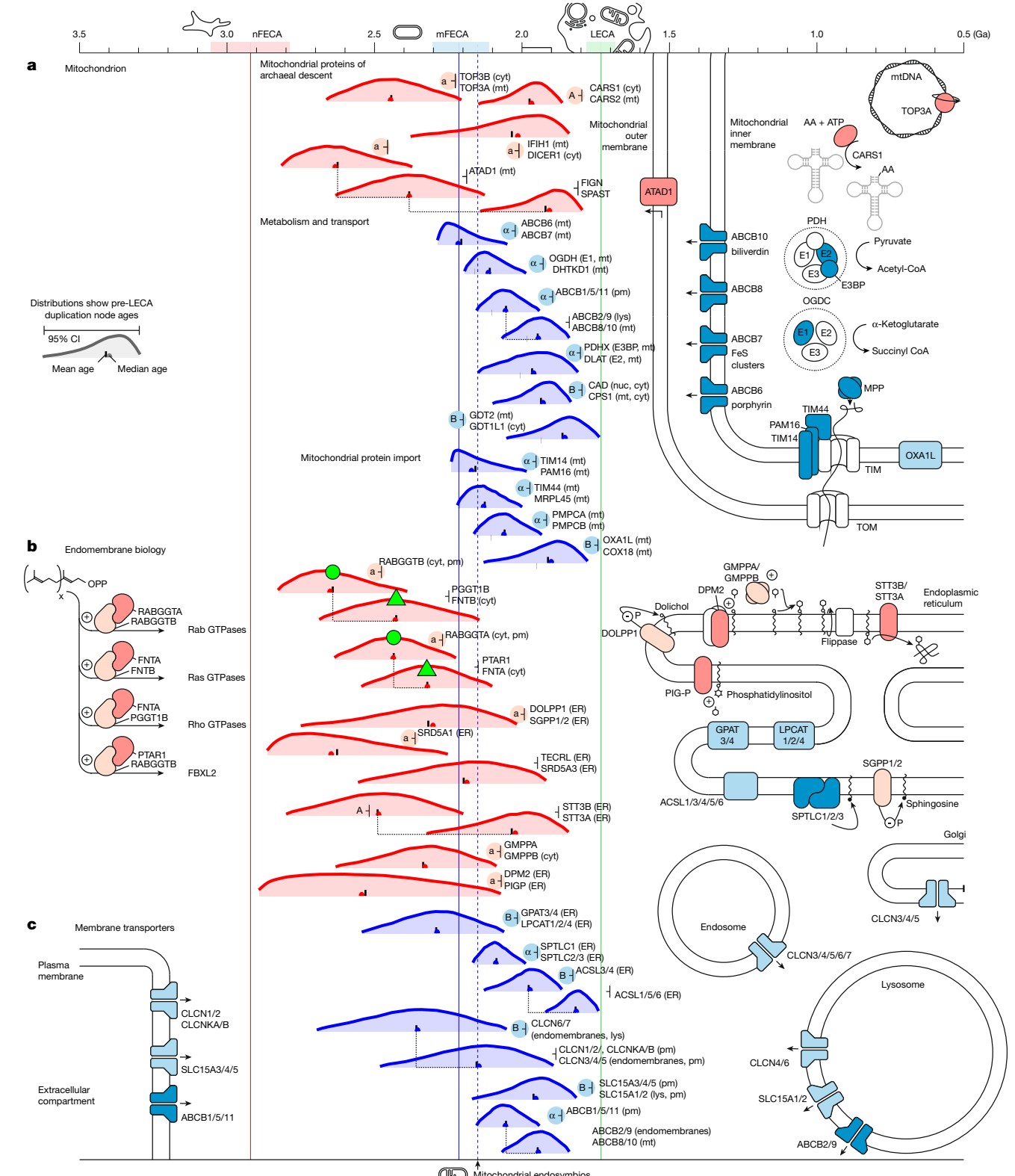

**Fig. 4 | Pre-LECA duplications in the mitochondrion and in endomembrane biology.** In contrast to the systems shown in Figs. 2 and 3, these systems have many bacterial origin pre-LECA duplications. **a**, Multiple gene families of alphaproteobacterial origin are seen to duplicate within 200 Myr of mFECA; we hypothesize that the onset of these duplications corresponds to the time of mitochondrial endosymbiosis. **b**, Membrane biogenesis involves families of both bacterial and archaeal descent, including paralogues of archaeal proteins that now function in bacterial membrane lipid biosynthesis. From this and other evidence, we infer an archaeal origin for the endomembrane system

that was later elaborated with gene products of bacterial descent. **c**, The endolysosomal system forms the digestive and recycling compartments of the eukaryotic cell. Specification of these compartments can be inferred from trafficking proteins (Fig. 3), and from the ages of their membrane transporter families we infer that this functional specialization emerged at the same time. Green shapes connect duplication events expected to have a similar age. Symbols next to each gene family indicate lineage of descent (a, Asgard; A, other archaea; α, alphaproteobacteria; B, other bacteria). AA, amino acid.

The duplication of RNA polymerase (pol) complexes into RNA pol I, II and III is consistent with changes to genomic organization more than 2.4 Ga. In modern eukaryotes, RNA pol I is localized to the nucleolus and transcribes ribosomal RNAs, RNA pol II transcribes mRNAs and RNA pol III transcribes tRNAs, 5S rRNAs and U6 snRNA, whereas the ancestral archaeal RNA polymerase complex performed all transcription. The divergence between RNA pol I and the others provides a minimum age for the development of a RNA pol I transcribed nucleolar compartment. The divergence of RNA pol III necessary for nuclear localized transcription of U6 snRNA and its divergence (2.73–2.32 Ga) is consistent with the development of the spliceosome. Since RNA polymerase complexes consist of multiple subunits, 3 duplications (Fig. 2b, green circles) testify to this age, falling in the range 2.91–2.34 Ga. The duplication ages overlap extensively, consistent with the hypothesis of synchronous duplication. We were also able to date several other gene families where, after gene duplication, one of the two paralogues today are targeted to the nucleus or nucleolus; these include nucleolar NHP2 (also known as SNU13) (Supplementary Fig. 3, 2.86–2.29 Ga), COPS1, COPS3, COPS5, COPS6 and COPS7 components of the COP9 signalosome (Supplementary Fig. 5, green circles), nuclear ACTL6 (Fig. 2a, 2.72–2.06 Ga), and RSL24D1 and EFTUD2 (Supplementary Fig. 6, 2.75–2.08 Ga and 2.56–2.18 Ga, respectively). On the assumption that the change in localization occurred after gene duplication, the age of these duplications provides a maximum estimate of the time at which each paralogue became targeted to the nucleus. The identified duplications fall in the period 2.86–2.08 Ga, consistent with continued elaboration of a nucleus from that point forward.

## The mitochondrion

Although our analysis identified fewer pre-LECA duplicates of bacterial origin than of archaeal origin, duplication ages for several families help to constrain the timescale for mitochondrial acquisition and integration. We identified three alphaproteobacterial gene family duplications that occurred between 2.30–1.97 Ga involved with the mitochondrial preprotein import system (TIM14 and PAM16: 2.24–1.99 Ga; PMPCA and PMPCB: 2.16–1.94 Ga; TIM44 and MRPL45: 2.22–2.00 Ga; Fig. 4a). As the respective proteins function in the inner mitochondrial membrane and establish access to the mitochondrial matrix, these duplication ages provide maximum constraints on the initial establishment of the interface between host and symbiont. Paralogues from early gene family duplications that are localized exclusively to the mitochondrion, and are targeted by preprotein import system signal sequences (the matrix enzymes DHTKD1 and OGDH: 2.19–1.99 Ga; PDHX and DLAT: 2.14–1.81 Ga; Fig. 4a), provide additional evidence for the early development of the preprotein import system in mitochondrial endosymbiosis. Mitochondria have an essential role in producing iron–sulfur clusters for the eukaryotic cell as a whole, and the inner mitochondrial membrane transporter ABCB7, which is involved in iron–sulfur cluster export from the mitochondrion, originated in a gene duplication (2.28–2.05 Ga). Prokaryotic homologues of this gene family have been characterized as heavy metal efflux transporters[35], and the pre-LECA eukaryotic paralogue ABCB6 is separately involved in haem export. This suggests that the functional specialization of ABCB7 was established after endosymbiosis, probably early in the process of mitochondrial integration.

Pre-LECA duplicates of archaeal origin also provide insight into the timing of mitochondrial integration in cases where one descendant paralogue was targeted to the organelle. We identify four gene family paralogues of archaeal descent whose other gene family members occupy different compartments (ATAD1, 2.63–2.13 Ga; TOP3A, 2.65–2.18 Ga; IFIH1, 2.42–1.99 Ga; CARS1, 2.09–1.83 Ga; Fig. 4a) in the interval 2.65–1.83 Ga. These proteins are another test of our approach, as we would not expect their duplication ages to be much older than mitochondrial endosymbiosis.

## Discussion

Previous analyses of pre-LECA gene duplications have delivered contradictory conclusions, suggesting either that genes of archaeal[2] or bacterial[3] ancestry accumulated more duplications along their respective eukaryotic stems. Our analysis focuses on pre-LECA duplications involved in eukaryotic apomorphies for which we were able to obtain alignments and gene family trees of sufficient quality for molecular clock analyses. We found many more genes of archaeal than bacterial origin with those qualities, consistent with previous work showing that eukaryotic genes of archaeal origin are in general more highly conserved[36], although this pattern may simply reflect a greater number of gene duplications on the archaeal versus alphaproteobacterial eukaryotic stem lineages.

### Comparing molecular clocks to other methods

Previous studies have used gene tree branch lengths, expressed in number of substitutions per site and normalized by relative evolutionary rates post-LECA, to infer the relative timing of events during eukaryogenesis, finding support for a relatively late mitochondrial endosymbiosis[2,12]. Relaxed clock methods provide crucial additional information for investigating these questions. First, clock methods link sequence divergence to the geological record, constraining the timing of key steps in eukaryogenesis to absolute time and, therefore, environmental context. Second, clock models provide a more flexible way to model variation in evolutionary rate through time, based on all of the available calibrations and sequence data—although we acknowledge that the real pattern of rate heterogeneity during eukaryogenesis and its associated HGT and duplications is likely to be more complex than captured by any current methodology. Finally, the Bayesian relaxed clock framework provides a natural way to propagate time information from the dated species tree—estimated using more calibrations and sequence data—to the individual gene trees, greatly ameliorating the difficulties resulting from the limited signal of short single-gene alignments. We note that this hierarchical framework implies that the inferred ages of gene duplications are informed by the ages of nodes on the dated species tree. For example, the inference that mFECA is younger than nFECA (Fig. 1) supports younger ages among duplications of alphaproteobacterial origin than those of Asgard origin, although sensitivity analyses demonstrate that this conclusion is robust to substantial variation in species tree ages (Supplementary Note 3).

### Timing eukaryogenesis

The timescale of lineage divergence in which we estimated the timing of gene duplications is broadly consistent with previous molecular dating analyses (Fig. 5a). Our estimates for the age of nFECA at 3.05–2.79 Ga and mFECA at 2.37–2.13 Ga are among the oldest, compared to a range of 2.90–2.09 Ga and 2.70–0.91 Ga, respectively, from previous studies[19,37–40]. Our 1.80–1.67 Ga estimate for LECA falls within the 2.39–0.95 Ga range of previous divergence time estimates[19,37–43]. These timescales are all based on relaxed molecular clock methods, but their differences reflect their underpinning data (sequence data), assumptions (clock model) and the nature of the calibrations used to disambiguate rates and times. Our estimates are closest to timescales that are based, as here, on analyses that used topology-based calibrations[19,38,40]. These enforce across-tree relative age constraints that reflect donor–recipient HGT and endosymbiosis events, spreading the limited temporal information from traditional node calibrations across the tree. As such, these timescales might be considered among the most realistic and, among them, our study has used among the most calibration constraints and sequence data. Our timescale infers a rapid radiation of the eukaryotic supergroups within about 300 million years (Myr) of LECA, consistent with the 'Big Bang' hypothesis of crown eukaryote diversification[44].

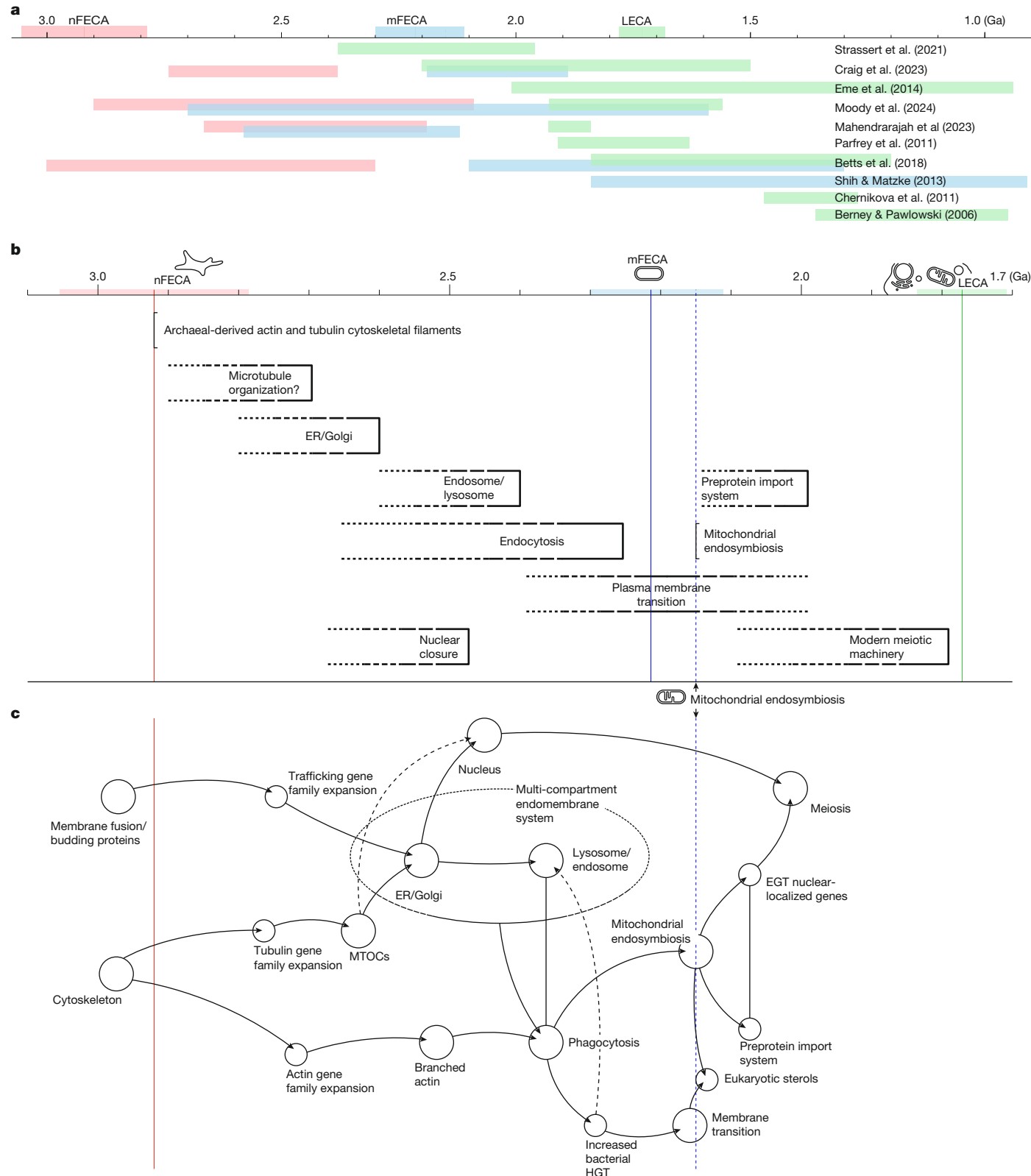

**Fig. 5 | Timeline of development for eukaryotic key apomorphies. a**, Our time-resolved species tree enables us to set a timeline for eukaryogenesis. Compared with other studies, our dates for nFECA and mFECA are among the oldest, whereas our date for LECA is intermediate[19,37–40,42,43,69–71]. **b**, Based on duplications in specific eukaryotic systems, we suggest a timeline for the emergence of these features. Vertical lines are suggested minimum limits for the emergence of features, and dashed horizontal lines denote the period of time for possible development and emergence. **c**, A tentative model that considers the interdependency of these features (arrowheads imply dependency; lines without arrowheads imply co-emergence but with as yet undetermined order). Data in **a**,**b** are aligned to the time axis, whereas in **c**, the nodes are grouped in relation to the nFECA and mitochondrial endosymbiosis boundaries. EGT, endosymbiotic gene transfer; ER, endoplasmic reticulum; MTOC, microtubule-organizing centre.

Given that our analysis is constrained by geologic evidence, it is pertinent to reflect on what aspect of eukaryote diversification the geologic record represents, not least because so little of it is used in calibration. Sterane records from the Neoarchaean Fortescue Supergroup of Australia[45] have been attributed to contamination[46] and, indeed, pre-Proterozoic biomarker records are generally considered questionable[47]. Large vesicles, compatible with (but not deterministic of) eukaryote affinity, are known from the Mesoarchean Moodies Group of South Africa[48]. Otherwise, the oldest widely accepted fossil eukaryotes are *Dictyosphaera*, *Shuiyousphaeridium*, *Tappania* and *Valeria* from the late Palaeoproterozoic (around 1.78 Ga) McDermott Formation of the Northern Territory, Australia[49] and the (approximately 1.64 Ga) Changcheng and Ruyang groups of North China[50–55], though their eukaryote affinity is based largely on inference of an actin cytoskeleton which evolved among archaeal ancestors[22]. *Qingshania*, also from the Chuanlinggou Formation, has a greater claim on eukaryote affinity, interpreted as a multicellular archaeplastid and, therefore, a late Palaeoproterozoic (approximately 1.63 Ga) crown eukaryote[56]. Otherwise, there is clear evidence of archaeplastids from the latest Mesoproterozoic[57] and earliest Neoproterozoic[58], among others[59], and possible Amorphea from the latest Mesoproterozoic[60]. Given the sparse nature of the fossil record through the Archaean to much of the Mesoproterozoic, we should not anticipate that these oldest records approximate clade age, but there is good fossil evidence for archaeplastids and, therefore, crown eukaryotes having diverged deep in the Proterozoic, as our timescale suggests. Although some of these records may be of metabolically active cells[61], the majority are cysts compatible with the prevalence of encystment among crown eukaryotes, the challenging environmental redox conditions that prevailed during eukaryogenesis and, evidently, crown eukaryote diversification.

There has been considerable debate about the environmental context of eukaryogenesis and of eukaryote diversification. It has long been argued that oxygenation of the biosphere, the Great Oxidation Event[62] (GOE; 2.43–2.22 Ga), was an environmental driver underpinning mitochondrial endosymbiosis and the origin of eukaryotes and, indeed, our evolutionary timeline precludes a syntrophic association between nFECA and mFECA prior to the GOE. However, the formative stages of eukaryogenesis are likely to have taken place under anoxia or hypoxia, as oxic conditions were limited to the surface waters of the Earth's oceans for much of the Proterozoic[8,9]. Specifically, our analyses point to an origin among archaea in the late Archaean, mitochondrial endosymbiosis almost coincident with the GOE, and diversification of crown eukaryotes in the late Palaeoproterozoic.

## Testing hypotheses of eukaryogenesis

From dated duplications in genes underpinning eukaryotic features, we suggest a model (Fig. 5b) in which the archaeal host progressively complexified through the sequential evolution of various characters before mitochondrial endosymbiosis. Our data suggest that duplications in archaeal membrane and cytoskeletal protein families first developed internal membrane compartments with a role in membrane biogenesis, and later in combination with cytoskeletal development and endocytosis. Although this does not imply that a host cell of crown eukaryote size and system-complexity existed before mitochondrial endosymbiosis, our results suggest that the host had the prerequisites for nuclear compartmentalization (development of the nuclear localization system), possessed an endomembrane system (from the diversification of compartment-specific vesicle trafficking proteins) and had an evolved cytoskeleton (with branched actin-specific paralogues) capable of endocytosis before mitochondrial endosymbiosis. Bacterial genes from the mitochondrial endosymbiont and other sources seem to have driven the membrane transition and further development of the endolysosomal system and to have added to existing nuclear processes such as DNA repair and gene regulation and new ones such as meiosis (Supplementary Discussion 1 and Supplementary Fig. 4).

The relative timing of mitochondrial endosymbiosis stands out as the model-defining feature of contemporary hypotheses of eukaryogenesis. In mitochondria-early models, such as the hydrogen hypothesis[5], the mitochondrial acquisition was the first step in eukaryotic evolution, and is suggested to have provided the energy needed to create a large cell and the eukaryotic apomorphies it supported[63]. By contrast, the phagocytosing archaeon[33], syntrophy[6,7] and serial endosymbiosis hypotheses[2,12] can be defined as 'mitochondria-late', since they defer mitochondrial acquisition until after the majority of other eukaryotic apomorphies and, in the case of syntrophy and serial endosymbiosis hypotheses, after the interaction with other bacterial partners. Other models such as the reverse-flow[30], inside-out[29] and E³ hypotheses[25] can be considered 'mitochondria intermediate'[1], as they do not make strong assumptions about when mitochondrial acquisition happened during eukaryogenesis.

Contemporary hypotheses of eukaryogenesis are principally evolutionary scenarios based on arguments of plausibility that serve to fill the large gaps in evidence between the living lineages from which data have been derived. In particular, Asgard relatives of eukaryotes encode various eukaryotic signature proteins, some of which have been shown to functionally resemble eukaryotic homologues[23]. Consistent with this genetic complement, the first cultivated Asgard archaea display remarkable cell biological features such as cytoskeleton-supported protrusions[25]. Furthermore, although fluorescence imaging indicates spatial separation of the genome from protein synthesis[64], in terms of overall size and complexity, the Asgard archaea cultivated so far resemble typical prokaryotes[25]. We have sought to bridge these gaps in our understanding of eukaryogenesis by establishing the timing and sequence of evolutionary innovations between the living lineages. Our results are not fully compatible with any of the competing evolutionary scenarios; many of the gene duplications that underpin eukaryotic apomorphies had already occurred in the archaeal nuclear lineage before the divergence of the mitochondrial endosymbiont from its closest sampled relatives, and so argue against mitochondria-early scenarios. Whereas several hypotheses have proposed that the development of the nucleus was a response to mitochondrial endosymbiosis[29], the results of our analyses suggest that nuclear development was underway before endosymbiosis. Other possible selective drivers for the evolution of the nucleus include protection of the nuclear genome[65], a means of preventing premature translation of non-spliced pre-mRNAs[29,66], or defence against viral infections[67]. The reverse-flow[30], inside-out[29] and E³ hypotheses[25] all suggest early cytoskeletal development in the archaeal host, consistent with our observations and the gene repertoires of extant Asgard archaea. The inside-out hypothesis goes further, possibly supporting the development of a nuclear localization mechanism before endosymbiosis. The phagocytosing archaeon hypothesis[33] also suggests a complex archaeal host, but differs from the other models in suggesting phagocytosis as the cell biological mechanism by which the mitochondrion was internalized. Our dating analyses suggest that the endolysosomal system evolved contemporaneously with the earliest alphaproteobacterial gene duplications (and therefore mitochondrial endosymbiosis), but we cannot discriminate which came first. Consequently, our results do not allow us to discriminate between the origin of phagocytosis and the timing of mitochondrial endosymbiosis, because the phagocytosing-archeon and mitochondrial-intermediate scenarios lack sufficient detail on the ordering of acquisition of the diversity of eukaryote novelties. Nevertheless, our results argue against the serial endosymbiosis hypothesis because the duplication ages of the genes implicated suggest that the majority of non-alphaproteobacterial genes began to duplicate after mitochondrial acquisition. Together, our results are most consistent with hypotheses that propose a complex host capable of endocytosis entering into a symbiotic relationship with the mitochondrion. We characterize models that agree with this order of events as complexified archaeon, late-mitochondrion (CALM) models of eukaryogenesis. Although it has been argued that the alphaproteobacterial endosymbiont may have

been an energetic requirement for the evolution of eukaryotic complexity[63], our results are instead consistent with views in which the early development of eukaryotic cell complexity was not dependent upon a mitochondrial partner, in agreement with recent theoretical work[68].

## Conclusions

Debate over the process of eukaryogenesis has been poorly constrained because there are no extant lineages representative of the component steps in building a eukaryotic cell. However, we have gleaned insights by establishing the ages of genes implicated in eukaryote novelties that have duplicated along the eukaryote stem lineages. Our results indicate that the process of eukaryogenesis began in the Mesoarchaean with divergence from archaeal relatives, involved acquisition of a mitochondrion in the early Palaeoproterozoic and culminated in LECA in the late Palaeoproterozoic. Furthermore, our data suggest that the host cell was already equipped with a cytoskeleton, endomembrane system and a nucleus (or protonucleus), or at the least with the physical separation of translation and transcription, prior to mitochondrial endosymbiosis. Our inferences on the relative timing of steps within eukaryogenesis do not support the early acquisition of mitochondria, but are compatible with most mitochondrial-intermediate and mitochondrial-late hypotheses. Mitochondrial endosymbiosis is inferred to have occurred shortly after the GOE. Given that Proterozoic Earth's oceans would have remained largely anoxic for more than a billion years, our inferences are compatible with eukaryogenesis models that invoke syntrophic interactions as a key driver underlying the symbiosis between the archaeal host and bacterial endosymbiont.

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

# Methods

Additional detail about the methods is given in the Supplementary Information and Supplementary Notes 1–4.

## Source data and processing

All analyses were conducted in Linux within a conda environment. Tree visualizations were produced in TreeViewer[72] and all other plots using Python Seaborn or Matplotlib, all programs and their versions are listed in Supplementary Table 1. Species data were obtained either from EukProt[73] or from NCBI. We relied on gene predictions when present and otherwise predicted open reading frames using TransDecoder (Supplementary Note 1 and Supplementary Table 2). Protein domain hidden Markov models (HMMs) were taken from PFAM[74] or PNTHR[75], or generated from curated reference proteins. Approximately 5,000 HMMs were used in our initial survey chosen based on their widespread occurrence in eukaryotes, and functionally annotated representatives in model organisms.

## Identifying candidate gene families

Candidate pre-LECA duplications were identified with the domain origins (DO) pipeline, which consists of the following steps: sequence retrieval (HMMER, v.3.3.2[76]), filtering and aligning (MAFFT, v.7.508[77]) these sequences, building new HMMs for iterating this search, clustering the results (MCL, v.22-282[78]), selecting representatives to produce a representative tree, and traversing this tree to identify the origin of eukaryotic clades. An extended description of this pipeline is given in Supplementary Note 1, a flowchart of its operation and outputs can be found in Supplementary Figs. 7–11. Summary trees from this pipeline were manually inspected before being carried further as candidates.

## Generating single gene trees

Identified gene families were further examined and subjected to molecular dating using a second pipeline—the domain analysis (DA) pipeline (Supplementary Note 1; flow diagram in Supplementary Fig. 12). The pipeline consists of two phases, the first phase prepares an optimal maximum likelihood tree from the input sequences, and the second phase time resolves this tree with MCMCTree (v.4.10.0[79,80]). The first phase performs the following principal steps; taxa reduction, deduplication, coarse removal of poorly fitting sequences, alignment maximization (MaxAlign, v.1.0[81]), alignment trimming (BMGE, v.1.12[82]), tree generation (IQ-tree and ModelFinder, v.2.2.0.3[83]) and species tree reconciliation. The first stage of this pipeline was used to prepare marker genes for the species tree, and both phases for the time resolution of pre-LECA duplications.

## Time-resolved phylogeny

An expansion on the species tree construction is given in Supplementary Note 2.

**Species tree topology.** Our goal was to produce a species tree in which all eukaryotic species were represented at least twice so that cross bracing could be applied (as in ref. 38). This requires duplicate LECA nodes and descendant eukaryote subtrees. We identified 62 candidate marker gene trees (Supplementary Table 3), which had eukaryotic descent from either alphaproteobacteria or Asgard archaea. Of these, 25 had gene tree nodes corresponding to the last universal common ancestor (with a combined length of 6,216 aligned amino acids), 40 with the last archaeal common ancestor (9,283 positions), and 43 with the last bacterial common ancestor (11,716 positions). From this set we produced concatenates using a custom script (Supplementary Note 2) for the whole time-resolved species tree (15,761 positions) as well as subset-specific concatenate alignments to explore prokaryotic and eukaryotic root placement within Archaea and Bacteria. Maximum likelihood trees using IQ-Tree were then constructed and taxa displaying discordant placements were manually identified and removed; this process was iterated until a largely concordant tree structure emerged. To arrive at a final species tree topology, several alternate topologies for the eukaryote and prokaryote roots were compared and tested in IQ-Tree with the concatenated sequence data (Supplementary Note 2).

**Sensitivity analyses.** We extended our topology tests to investigate the impact of species tree topology and calibration on time resolution. These tests were focused on the impact in two main areas, the age of deep nodes in the tree (LACA, LBCA and LUCA), and the robustness of the eukaryogenesis timeline (mFECA, nFECA and LECA), the results of these tests is given in Supplementary Note 3 and Supplementary Tables 4 and 5. In general, we found the ages of the FECA ages and their relative branch lengths to be robust to the uncertainty of deeper nodes in the tree (LACA and LBCA), changes to eukaryote species tree topology did slightly affect LECA ages but not FECA ages or relative FECA stem length.

**Time resolution with MCMCTree.** Time-resolved species trees were modified for MCMCTree[79] by removing branch lengths and support values. Uniform fossil calibrations were then applied (Supplementary Note 4 and Supplementary Table 6). The duplications option in PAML was used to formally cross-brace species splits, and we applied this for repeated fossil calibration nodes (18 instances), as well as congruent species splits (88 instances). MCMCTree was then run using the LG + F + G5 substitution matrix[84], 10,000 chains of burn in, and a further 20,000 chains sampled at a sample frequency of 1 in 2. This was repeated in quadruplicate and checked for consistency and convergence.

## Dating pre-LECA duplications

**Species tree priors.** In the sequential Bayesian approach, the time-resolved species tree is used to produce new priors which are then applied to single gene trees. Pooled MCMCTree runs were used to produce skew-T distributions and maximum age constraints for each speciation node, where these nodes exist on the single gene trees the constraint is mapped onto the node, where a constraint is present more than once, it is cross-braced. We also place constraints on the age of the duplications (between LECA and archaeal/bacterial FECAs) where the origin of the gene is Asgard archaeal (or sister to the TACK superphylum) or alphaproteobacterial (or sister to) in origin. The placement of calibrations is shown in the domain analysis reports.

**MCMCTree on single gene trees.** The analysis of individual gene trees follows a similar methodology to that of the species tree and uses the second phase of the DA pipeline. Maximum likelihood trees generated using IQ-tree were rooted and converted into a format compatible with MCMCTree. Calibration priors were algorithmically applied to species splits and groups and finalized through manual inspection and annotation of critical nodes (for example, LECA or duplications). MCMCTree was then run in quadruplicate (same burnin and sample as species tree) and results examined for consistency and convergence in the trace plots produced in the single gene tree reports. It is from these analyses we obtain the pre-LECA duplication dates for our studied domains. The investigated duplicated domains are given in Supplementary Table 7.

## Gene function annotation

Localization and function information for LECA groups was assigned on the basis of human reference proteins in the NIH gene database[85]. Whenever multiple localizations were reported, we have preferred the compartment where the gene is described as 'active'; similarly, we have preferred the human protein names to represent these groups. We noticed that different domain HMMs recovered the same set of human proteins as a result of proteins having multiple domains or domains being related to each other). We identified these cases from the duplicated human reference proteins. A consequence of this is

that we report fewer total duplications, but we also note that in these repeat analyses the ages for duplications were found to be largely in agreement.

## Reporting summary

Further information on research design is available in the Nature Portfolio Reporting Summary linked to this article.

## Data availability

The dataset generated in the course of this study is available at the University of Bristol data repository (data.bris; https://data.bris.ac.uk/data/) at https://doi.org/10.5523/bris.tjfgfs0kmr532t2gvqpcdbmto.

## Code availability

The code used to generate our results is provided with the data stored at the University of Bristol research data repository with the identifier https://doi.org/10.5523/bris.tjfgfs0kmr532t2gvqpcdbmto.

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

**Acknowledgements** This work was supported by a grant from the Gordon and Betty Moore Foundation (GBMF9741 to T.A.W., P.C.J.D., D.P., G.J.S. and A.S.). This study was conducted with the support of John Templeton Foundation (grants 62220 and 63451 to T.A.W., P.C.J.D., D.P., G.J.S. and A.S.; the opinions expressed in this publication are those of the authors and do not necessarily reflect the views of the John Templeton Foundation), the Biotechnology and Biological Sciences Research Council (BB/T012773/1 and BB/Y003624/1 to P.C.J.D.) and the Leverhulme Trust (RF-2022-167 to P.C.J.D.; and RPG-2024-030 to D.P.). A.S. has also received funding from the European Research Council (ERC) under the European Union's Horizon 2020 research and innovation programme (grant agreement no. 947317, ASymbEL).

**Author contributions** P.C.J.D., D.P. and T.A.W. conceived the study. C.J.K. designed and conducted the investigation. C.J.K. created software used in the work. P.C.J.D., D.P., T.A.W., C.J.K., A.S. and G.J.S. contributed to analysis and discussion, and all authors contributed to writing and preparing the paper.

**Competing interests** The authors declare no competing interests.

**Additional information**
**Correspondence and requests for materials** should be addressed to Christopher J. Kay, Davide Pisani, Tom A. Williams or Philip C. J. Donoghue.

# Reporting Summary

## Statistics

For all statistical analyses, confirm that the following items are present in the figure legend, table legend, main text, or Methods section.

| n/a | Confirmed | |
|---|---|---|
| ☐ | ☒ | The exact sample size (*n*) for each experimental group/condition, given as a discrete number and unit of measurement |
| ☒ | ☐ | A statement on whether measurements were taken from distinct samples or whether the same sample was measured repeatedly |
| ☒ | ☐ | The statistical test(s) used AND whether they are one- or two-sided *Only common tests should be described solely by name; describe more complex techniques in the Methods section.* |
| ☒ | ☐ | A description of all covariates tested |
| ☒ | ☐ | A description of any assumptions or corrections, such as tests of normality and adjustment for multiple comparisons |
| ☒ | ☐ | A full description of the statistical parameters including central tendency (e.g. means) or other basic estimates (e.g. regression coefficient) AND variation (e.g. standard deviation) or associated estimates of uncertainty (e.g. confidence intervals) |
| ☒ | ☐ | For null hypothesis testing, the test statistic (e.g. *F*, *t*, *r*) with confidence intervals, effect sizes, degrees of freedom and *P* value noted *Give P values as exact values whenever suitable.* |
| ☐ | ☒ | For Bayesian analysis, information on the choice of priors and Markov chain Monte Carlo settings |
| ☒ | ☐ | For hierarchical and complex designs, identification of the appropriate level for tests and full reporting of outcomes |
| ☒ | ☐ | Estimates of effect sizes (e.g. Cohen's *d*, Pearson's *r*), indicating how they were calculated |

*Our web collection on statistics for biologists contains articles on many of the points above.*

## Software and code

Policy information about availability of computer code

| Data collection | All programs used in this study are listed with their version numbers in Supplementary Table 1, a full description of the novel software created in this study is expanded upon in Supplementary Notes 1, and in the Supplementary Material. Descriptions of how to run the programs, their requirements, sample software environments and sample data are provided in the Supplementary Material, which is stored on the public archive Zenodo. |
|---|---|
| Data analysis | As above. |

For manuscripts utilizing custom algorithms or software that are central to the research but not yet described in published literature, software must be made available to editors and reviewers. We strongly encourage code deposition in a community repository (e.g. GitHub). See the Nature Portfolio guidelines for submitting code & software for further information.

## Data

Policy information about availability of data

All manuscripts must include a data availability statement. This statement should provide the following information, where applicable:
- Accession codes, unique identifiers, or web links for publicly available datasets
- A description of any restrictions on data availability
- For clinical datasets or third party data, please ensure that the statement adheres to our policy

All source data (genomes / proteomes) was obtained from public sources, and listed in the Supplementary Table 2, a complete copy of all the proteomes used in

this study, the analysed gene trees and the time resolved analyses of these gene trees is provided in the Supplementary Material, sufficient to reproduce our findings. All this material is stored on the public archive Zenodo.

# Research involving human participants, their data, or biological material

Policy information about studies with [human participants or human data](). See also policy information about [sex, gender (identity/presentation), and sexual orientation]() and [race, ethnicity and racism]().

| | |
|---|---|
| Reporting on sex and gender | n/a |
| Reporting on race, ethnicity, or other socially relevant groupings | n/a |
| Population characteristics | n/a |
| Recruitment | n/a |
| Ethics oversight | n/a |

Note that full information on the approval of the study protocol must also be provided in the manuscript.

# Field-specific reporting

Please select the one below that is the best fit for your research. If you are not sure, read the appropriate sections before making your selection.

☐ Life sciences    ☐ Behavioural & social sciences    ☒ Ecological, evolutionary & environmental sciences

For a reference copy of the document with all sections, see [nature.com/documents/nr-reporting-summary-flat.pdf]()

# Ecological, evolutionary & environmental sciences study design

All studies must disclose on these points even when the disclosure is negative.

| | |
|---|---|
| Study description | This study is a time resolved phylogenetic analysis of eukaryotic gene families which have origins deeper than that of the eukaryotic common ancestor. |
| Research sample | This study examined a collection of publicly deposited proteins and genomes, from which we sampled a set of ~500 taxa, with an aim to capture the diversity of all living things. |
| Sampling strategy | as above |
| Data collection | Data collection was performed by Christopher Kay |
| Timing and spatial scale | n/a |
| Data exclusions | n/a |
| Reproducibility | All data necessary to reproduce our results are included in the Supplementary Material |
| Randomization | n/a |
| Blinding | n/a |

Did the study involve field work?    ☐ Yes    ☒ No

# Reporting for specific materials, systems and methods

We require information from authors about some types of materials, experimental systems and methods used in many studies. Here, indicate whether each material, system or method listed is relevant to your study. If you are not sure if a list item applies to your research, read the appropriate section before selecting a response.

## Materials & experimental systems

| n/a | Involved in the study |
|-----|------------------------|
| ☒ ☐ | Antibodies |
| ☒ ☐ | Eukaryotic cell lines |
| ☒ ☐ | Palaeontology and archaeology |
| ☒ ☐ | Animals and other organisms |
| ☒ ☐ | Clinical data |
| ☒ ☐ | Dual use research of concern |
| ☒ ☐ | Plants |

## Methods

| n/a | Involved in the study |
|-----|------------------------|
| ☒ ☐ | ChIP-seq |
| ☒ ☐ | Flow cytometry |
| ☒ ☐ | MRI-based neuroimaging |

## Plants

| | |
|---|---|
| Seed stocks | n/a |
| Novel plant genotypes | n/a |
| Authentication | n/a |

