## [Peer Review File · Nature]

Dated gene duplications elucidate the evolutionary assembly of eukaryotes

Corresponding Author: Professor Philip Donoghue

Version 0:

Reviewer comments:

Referee #1

(Remarks to the Author)

Peer review report for Kay et al.

2025-03-07178 "Dated gene duplications elucidate the evolutionary assembly of the eukaryotic cell"

Summary

In their study 'Dated gene duplications elucidate the evolutionary assembly of the eukaryotic cell', Kay and colleagues use a geological record-anchored molecular clock to date key gene duplication events that occurred during the evolution of eukaryotes. While time estimates remain wide, this allows the authors to identify genes that likely duplicated early on in eukaryotic evolution (prior to the acquisition of the alphaproteobacterial ancestor of the mitochondrion); those that duplicated around the time of mitochondrial domestication; and those that duplicated later along the path to LECA.

In analyzing their findings, the authors identify duplication events that are consistent with the emergence of new functions which are seen as instances in which functionally related machinery duplicates at a similar time. These events can be used to constrain hypotheses about the likely path of eukaryogenesis.

Amongst this set of gene duplications, there are several that are notable, including the following:

i) The authors identify duplications in genes with both archaeal and bacterial lineages involved in embedding mitochondrial endosymbiosis. These include: MRPL45 – a mitochondrial ribosomal protein, CARS2 – an archaeal derived tRNA synthetase operating in mitochondria (could this be related to Fe-S formation?), and the protein import system TIM14. Importantly, all appear after the "great oxygenation event."

ii) DNA damage repair system evolution is striking because of its mix of early archaeal proteins and late bacterial-derived proteins. This analysis has implications for the dating of the evolution of meiosis, and suggests a new way to think about DNA damage-repair in human disease.

In a field that has too many hypotheses and not enough data, this paper marks a very important step forwards. For this reason, the data presented will interest a very wide audience of evolutionary and molecular cell biologists and should spawn research into the evolution of many systems whose emergence via a gene duplication is touched on in the paper.

Overall, the breadth and depth of the analysis carried out is impressive, the paper is well written, the Figures are relatively clear, the discussion is excellent, and the text is supported by a long, but very helpful supplement.

There are a couple of issues that we suggest be addressed prior to publication.

Major comments:

- The analyses are based on the use of a limited sample of Asgard genomes. In doing so, they have excluded

representatives of the Hodarchaeales. This seems like an unfortunate omission, since these Archaea have been suggested to be the closest archaeal relatives of eukaryotes. We would therefore strongly suggest that the authors consider including additional Asgard genomes in their analyses.

- The paper implies that there was a burst of gene duplication events in the nFECA lineage that was associated with the acquisition specific eukaryotic cell functions. There seems to have been no attempt to measure the change in rate, even though the paper states: “our analyses suggest that in the nuclear lineage the rate of duplicate gene fixation is even higher than in other Asgard.” More evidence to support this statement would be welcome.
- It would be useful on this point to include a more systematic analysis of duplication events that occurred earlier in the archaeal stem lineage (in Asgard, Heimdall and Hod lineages) that led to the divergence of homologs that retain functional differences in eukaryotes that are already included as subjects of the analysis in the paper. Examples of these could include: Sm proteins, ESCRTs, AAA-ATPases. Such an analysis would provide important context for the results presented in the paper. Some of these duplication and divergence events were likely important for the evolution of compartments prior to nFECA. While some Asgard duplications (e.g. tubulins) did not make it into the eukaryotic lineage, they could be used to provide evidence of selective pressure for duplication events in some protein families within the Asgard archaea/eukaryotes. It would also be useful for readers if the paper had a fuller discussion about how definitions of nFECA are likely to change as more Asgard lineages and other potential close eukaryotic relatives are identified over the coming years.
- While the Figures are great, they still need work. To help readers, Figures and/or their legends should have a key that explains everything shown – e.g. GREEN SHAPES. It is not enough that this is explained in the text. In addition, the names of all proteins in Figures and related text should be consistent and easy to find, e.g. TREX1 and CENP-A.
- “Compartment/organelle function” calling is dubious and should be avoided where possible. Instead of saying “vesicle trafficking” or “nucleus” it is much better to explain what the protein does or to state where it is localized in modern eukaryotes, and/or which other proteins it interacts with. This is especially true when we are considering the emergence of eukaryotic cell organization, since protein functions might emerge before the cell structure in which they act in extant eukaryotes. As an example of this, protein glycosylation occurs at the surface of archaea, but begins in the ER and is augmented in the Golgi in eukaryotes, so it’s better to state this as a protein activity related to secretion, not to an organelle.

Other examples:

ESCRT-III proteins, like CHMP4C, are membrane remodeling cytoskeletal proteins. There is very limited evidence to suggest that they play an important conserved role in kinetochore function.
Histones and SMC proteins regulate chromatin architecture whether or not there is a nucleus. And NLS sequences likely predate nuclei in many proteins where they may function to localize proteins to nucleic acids. The Exosome and Sm proteins are involved in RNA processing in both archaea and eukaryotes. There is also evidence for nucleolar-like activities in archaea (Sulfolobales) and perhaps a nucleolar-like structure. This is important to be clear about, since the emergence of splicing, something that is specific to eukaryotes and depends on the presence of a nucleus, needs to be thought about very carefully.

As part of this, Figures are easier to make sense of when proteins with related molecular functions are placed together as has been done for much of the paper. This could be fleshed out in a table or additional figures.

- i) AAA ATPases (Cdc48, PAN, ATAD1, FIGN/Spast, Vps4...)
- ii) Lipid metabolism
- iii) DNA repair
- iv) DNA polymerases
- v) Microtubule function (Tubulin, gamma Tubulin, Dynactin, HATs, HDACs).

Note that this type of analysis is especially informative when eukaryotic activities involve a combination of bacterial- and archaeal-derived components, e.g. in SFigure 12. In this sense, it’s worth highlighting the timeline in the evolution of i) Histones with tails (archaeal-derived), ii) Methyl transferases (bacterial-derived), and iii) HDACs (archaeal-derived). What about HATs?

- It would be good to include a fuller general discuss of how all the data taken together shed light on the relative timing of gene duplications associated with the acquisition of activities that are now confined to the ER, Golgi, nuclear envelope, plasma membrane, mitochondrial outer membrane, endosomes, and the lysosome (Rag GTPases / TOR complexes / V-ATPase etc).

The emergence of the following genes seems an important early marker of this: SGPP1/2 SRD5A1 2.86-2.26 Ga; DPM2/PIGP 2.89-2.08 Ga; STT3A/B 2.80-2.20 Ga).

Minor comments

- It would be good to be clearer in the main text about the potential function and source of bacterial genes in 1E that duplicated early.
- The key for Figure 1B refers to mFECA as 22.0Ga.
- Authors refer to both ‘Heimdallarchaeota’ and ‘Asgardarchaeota’ in the text. Authors should adhere to one naming system.

- In Figure 2A, it is odd to refer to cytoskeletal genes as “vesicle trafficking genes.” Just call them “cytoskeleton genes”.
- CHMP4C is an ESCRT-III homologue.
- Some author comments on the Supplemental text have not been addressed or deleted!
- METTL4, not “METL4”, belongs to a subclade of MT-A70.
- There is no evidence to show that cytoskeleton “pattern” protrusions in Asgard. All one can say for the moment is that they are present in Asgard cells and in their protrusions.
- It is not certain that phagocytosis requires branched actin. All it needs is binding, membrane spreading and scission. Scission is aided by actin and by Dynamin. Did the authors look at Dynamin duplications? Did the authors look at any actin regulators? Formins?
- Microtubules are not always localized around MTOCs. This is only really true of some animal cells. Like actin, microtubules need nucleators - gamma tubulins.
- The following sentence is confusing: “The duplication of RNA polymerase complexes into RNA pol I, II and III is also consistent with the development of the nuclear compartment >2.4 Ga.” Why does this follow?

(Remarks on code availability)

Referee #2

(Remarks to the Author)

I co-reviewed this manuscript with one of the reviewers who provided the listed reports.

(Remarks on code availability)

Referee #3

(Remarks to the Author)

Kay et al. have dated and studied duplicate gene families across prokaryotic and eukaryotic diversity to elucidate the relative timing of key events in eukaryogenesis – including the endosymbiotic origin of mitochondria. The result is a remarkable dataset that provides many insights into thorny issues that researchers in the field of eukaryotic evolution have been grappling with. Some of the conclusions are complementary to other papers, such as rejecting ‘mitochondria-early’ scenarios in favour of increased complexity of ‘eukaryotic’ molecular and cellular features in archaea. But I am aware of no study that has addressed so many different elements of eukaryogenesis in such a synthetic and insightful manner.

I can’t find fault with the timescale / relaxed clock approach the authors have taken. It is far from perfect, but the approach has already been established, is clearly described, and the caveats and limitations are adequately addressed. There are no doubt conflicting signals and biases in the underlying data that push specific results this way and that; but the strength of the paper is in the sheer volume of data analyzed such that larger trends can be inferred. It really is a remarkable study that will serve the field well.

It’s not a perfect paper. There are errors and deficiencies throughout (see below), and while the main text is manageable in isolation, the volume of information presented in the Supplementary Discussion / Notes is simply overwhelming. In some cases, it also seems redundant. See below for specific examples. I would encourage the authors to consider how much of the information in Supp info is truly necessary, and trim down accordingly.

Specific comments.

-Line 444: “CALM: Complexified Archaeon, Late Mitochondrion models of eukaryogenesis”. This is mentioned only once. I’m not sure the field would benefit from another acronym, in this case one that refers to a certain flavor of eukaryogenesis models. I don’t feel strongly about it, but I doubt it would be very useful.

-Redundant main text and supp info figures. Figure 2 in the main text (endomembrane system) has a counterpart: Supp Figure 15. They are different, but only slightly so. I can’t tell the difference between Figure 4 and Supp. Figures 17. Why not replace the former with the latter? It’s not clear to me why there is so much redundancy between main text figs 2-4 and the Supp figures.

-Figures in the Supplementary Notes – a good number of them cannot be read in the pdf file that I reviewed (e.g., Fig S7, S9). If they can’t be read when zoomed in, then there is no point in having them be a Supp figure).

-Line 93/94. Redundant with statement and abbreviations in the introduction (lines 67-69)

-LBCA used only twice in MS, on lines 104 and 107. Spell out for non-specialists.

-Line 119 – ‘total-groups’ = typo? If not, didn’t make sense to me in this context.

-Line 132: “Based on these considerations, we hypothesize that the initial accumulation of gene duplications in the mitochondrial lineage, after its divergence from Alphaproteobacteria, must reflect a shift to a more eukaryote-like population-genetic environment.” Pop gen factors certainly would have played a role in the likelihood of duplicates being fixed, but another factor is that proto-mitochondrial duplicates would have been exploring new biological / biochemical landscapes in the context of a proto-eukaryotic cell. Things like TIMs are mentioned, of many possible functions that would have been massively rejiggered during this tumultuous time of cell evolution, not to mention alpha-proteobacterial-derived genes / proteins that ended up having nothing to do with the mitochondrion. I guess what I am saying is that there is fixation and then there is functional divergence – which of course are related – but it’s possible for dups to be fixed with little / no functional divergence and this paper is all about eukaryotic diversification, not population genetics. So I suggest perhaps rethinking the language a bit at this point of the text to go more along with the main focus.

-Figure 1 and legend needs some work.

-plastid, not plasmid

-Part D is not visually distinct from parts C and E (maybe lower the ‘D’ so it is less seemingly part of the bottom of the histogram in C).

-Part E. What are the horizontal lines, exactly? They are the 19 non-alpha dups, right? Can they be labeled individually, or at least explicitly labeled as a set so that they at least can be seen as ‘other bacterial duplicates? Why do some have 1 light blue ticks at right end, but some have a second internal blue tick? None of this is explained in the legend at all.

-Finally, I’m not sure this figure needs the labels at the very bottom (nFECA, mFECA, etc.) to be labeled part F. They align with the icons and dates at the top of the right side of Figure 1 (i.e., right above the histogram in part C), but the images at the top don’t have their own label.

-To conclude: while the data in Figure 1 right are very interesting and informative, it takes a lot of staring to figure out what exactly is being shown. I encourage the authors to re-think the layout a bit.

Figure 2.

-If I am interpreting the data correctly, delta and epsilon tubulins seem to be older than the alpha/beta paralogs. Brief comment on the cell biological significance of this WRT microtubule evolution for centrioles, etc.

-Error in top right. The main part of the microtubule is a alpha-beta heterodimer, but the figure includes gamma in the dimer. This error also appears in Supp Figure S16.

-gene families of Asgard archaeal origin are shown with an ‘a’ in the figure but described with an ‘alpha’ symbol in the legend. This ‘a’ / ‘alpha’ issue applies to Figures 3 and 4 as well.

General comment – the ‘a’ / ‘alpha’ paralogs have clear asgard homologs and thus are special, more special than is obvious from the figure. Beyond sorting out the ‘a’ / ‘alpha’ issue, I would suggest making them stand out more in the figure. Perhaps with a colored circle with an ‘a’ / ‘alpha’ symbol inside it? There is already a ton of data in these figures, and the tiny ‘a’ / ‘alpha’ beside the paralog names almost goes unnoticed.

Figure 4.

-What does ‘Genes duplicated into the mitochondrion’ mean (top left)?

-The text refers to 4C, but there is no such part C labeled in the figure (it’s obviously the bottom part analyzing the CLCN paralogs involved in endosome / lysosome functions).

-Line 158: “While many of the genes that specify these structures are specific to eukaryotes, others descend from prokaryotes.” It’s a nit-picky point, but technically ALL the genes that specify “these structures” descend from prokaryotes, with the exception of cases of novo gene evolution from non-coding DNA. De novo gene evolution is now recognized as ‘a thing’, but I think a minor rephrase is needed to avoid confusion for non-specialists. Many and probably most ‘eukaryote-specific’ genes ultimately came from prokaryotic homologs but are no longer recognizable as such. 25 / 30 years ago we didn’t realize that prok. Homologs of tubulins, actins etc. can in fact be found, with the help of 3D structures, despite the extreme primary sequence divergence. Indeed, they are discussed in the very next paragraph.

-Line 215 – ER-localized proteins, not ER-localized genes

-Supp Info files and Supp Notes are out of order. The first SI file mentioned at the very beginning of results is Fig S10, the first Supp Note is S3.

-Line 300 – “We identify 4 gene family paralogs of archaeal descent, whose other gene family members occupy different compartments (ATAD1 2.63-2.13 Ga; TOP3A 2.6-2.2 Ga; IFIH1 2.37- 1.84 Ga; CARS1 2.14-1.87 Ga; Fig. 3A, Supplementary Discussion 1) in the interval 2.63-1.87 Ga.” You mean Figure 4, right?

-Discussion – GOE is described very late in the MS and not actually spelled out. This could be confusing to non-specialists, since based on context alone it could be assumed to be great oxygenation or great oxidation. So I would spell it out and have a look at this section of the text so that the main message to readers is clear and unambiguous.

(Remarks on code availability)

Referee #4

(Remarks to the Author)

A. Summary of the Key Results

The authors address a long-standing and fundamental question in evolutionary biology: the timing of key events during eukaryogenesis. This is a particularly difficult question because there are no intermediate stages and few fossils. Various approaches have been developed to answer this question, often focussing on gene duplication events. Here, the authors use a relaxed molecular clock approach applied to gene duplications and reconstruct a timeline that suggests that key cellular features (cytoskeleton, endomembrane system and nucleus) evolved prior to mitochondrial endosymbiosis. They propose that nFECA evolved between 3.1–2.8 Ga, mFECA between 2.4–2.2 Ga and LECA around 1.8–1.7 Ga.

B. Originality and Significance

This study is highly relevant and methodologically advanced. While the biological conclusions are consistent with previous work (e.g. Pittis & Gabaldón 2020; Vosseberg et al. 2021) already suggested substantial host complexity prior to mitochondrial acquisition, the authors reinforce these ideas with newly compiled datasets and careful dating of the evolutionary events. Explicitly highlighting the methodological novelties, such as specific improvements to the datasets or refinements to the analyses, would further enhance the originality of the study.

Overall, the study makes an important contribution by rigorously testing an important evolutionary hypothesis with an advanced methodology and a large dataset.

C. Data & Methodology: Validity of Approach, Quality of Data, Quality of Presentation

The methodology is well chosen, carefully explained and replicable. The selection of hallmark genes associated with important cellular systems provides a solid biological basis for the evolutionary interpretations. However, some clarifications are needed:

- It would be helpful to make clear how the dataset differs from previous studies, in particular, the selection criteria for the proteomes used and the genes used in the analyses.
- The term “downloaded proteomes” should be defined more precisely. Are they predicted proteins from complete genomes?

Were all available proteomes used or was filtering performed?

- The figures are very nice and informative and help a lot to convey the results.

D. Appropriate Use of Statistics and Treatment of Uncertainties

The statistic is appropriate and the methodology is applied with care. Nevertheless, further discussion of the assumptions and known limitations of molecular clock models, especially with respect to deep-time phylogenies, would be valuable. In particular, genes acquired through endosymbiosis or HGT might have different duplication rates compared to native archaeal genes, which could affect clock estimates even in a relaxed model. Addressing this point would increase confidence in the results.

E. Conclusions: Robustness, Validity, Reliability

The conclusions are robust and generally well supported. However, some points could be expanded:

- Are there specific functional categories associated with duplications after mitochondrial acquisition?
- Are there recognisable patterns by which archaeal genes duplicated before versus after mitochondrial acquisition?
- The energetic cost of complexification remains a major argument against a mitochondrial late scenario; this should be discussed further.

F. Suggested Improvements: Experiments, Data for Possible Revision

There is no major flaw that would require new experiments. However, a clearer description of a selection of data sets, a slightly expanding the discussion of uncertainties, and better highlighting of methodological innovations would improve the manuscript.

G. References: Appropriate Credit to Previous Work

The authors make appropriate reference to relevant previous studies.

H. Clarity and Context: Lucidity of Abstract/Summary, Appropriateness of Abstract, Introduction, and Conclusions

The manuscript is generally clear, logically organised and well written. A few minor points:

- The novelty of the results is somewhat overstated in the abstract.
- The differences to previous work should be emphasised more clearly.
- There is a small inconsistency in the dating results between sections (e.g. 3.1–2.8 Ga vs. 3.05–2.79 Ga for nFECA); consistency should be ensured.

Conclusion

This manuscript presents a compelling and carefully conducted study that makes a valuable contribution to the understanding of the evolutionary origins of eukaryotic complexity. To further enhance its impact, the authors should more clearly emphasise the methodological advances, expand the discussion of uncertainties and clarify some minor methodological details.

(Remarks on code availability)

Referee #5

(Remarks to the Author)

I have read this manuscript with great pleasure and interest. The authors set out to establish a chronology of the various events along the eukaryogenesis process by using dated gene trees. They succeed in calculating reliable node ages for various gene duplication events functionally linked to various eukaryogenesis-related events, and use this information to provide an absolute and relative chronology of the process. One of the most interesting aspects of the study is that the authors go well beyond aggregating node age distributions, examining the implications for each gene family in quite detail. In addition, the use of a sequential Bayesian approach to time-calibrate single gene trees using a reference species tree is (to my knowledge) a welcome and important innovation in the study of early eukaryotic evolution. Even if it is used mostly as a point of reference for gene-level analyses, the construction of a robustly-dated species tree that reflects the dual bacterial/archaeal ancestry of present-day eukaryotic genomes is, in itself, a major achievement.

The authors take great care in contrasting their findings with an array of previous hypotheses about the eukaryogenesis process, listing strong and weak compatibilities/incongruences with each of them in the Discussion section. This well-warranted caution does not preclude the authors from enunciating the fundamental characteristics of their eukaryogenesis model (early complexification, non-early mitochondrial acquisition...), and putting forward an accompanying acronym ("CALM"). The fact that the gene-level evolutionary histories supporting their results are readily available in well-annotated supplementary notes will greatly facilitate their examination by other researchers and prompt further research in this field. Yet, there are some parts of the manuscript that could have been prepared with more care. For example, figures are often referenced out of order (especially the abundant supplementary materials) and incompletely described (with explanations being scattered between in-figure legends, captions, main text, and supplementary notes), and the summary Methods in the main text, while welcome, are hastily written (e.g. referencing concepts that do not appear elsewhere in the text). I list some of these issues in the line-by-line comments below.

Other than that, the phylogenetic analyses here presented are of very high quality, the results ("The development of key eukaryotic traits", L156-304) are described clearly, and the discussion derived from them is rich and well-supported (L305-onwards).

Comments on the manuscript

L95: A brief description of how the 64 marker genes used in the species phylogeny were chosen would be most welcome at this point. Presumably, these include a mixture of genes whose eukaryotic homologs were inherited from archaea (nLECA branches) or bacteria (mLECA/pLECA branches), but that are otherwise shared among the various prokaryotic lineages, is that correct? The authors could also provide an indication of how many markers (and alignment positions?) fall within each category.

L98: The sensitivity analyses mentioned here are an important part of the phylogenetic analyses of the study. The rationale behind their importance and the basic insights derived from them should be summarised here. Also, the authors should be more specific about where to find this in the supplementary Methods (here and elsewhere in the paper). In this instance, the term "sensitivity analysis" does not appear in the main text methods.

L104: "To investigate the impact of this uncertainty in LBCA age..." — "LBCA" means last bacterial common ancestor, correct? The rationale for these additional sensitivity analyses is not clear in the main text, not the least because the phrase "this uncertainty" implies that this issue has already been touched upon. Why is this problem not relevant in the archaeal/nLECA portion of the tree? Is it related to the relative age of the LBCA/LACA nodes? Is it due to differences in the amount of alignment positions (or number of reliable markers) in either of these areas of the tree?

L116: The number of gene trees used in this study (136) is only a subset of all the eukaryotic gene families inherited from either bacteria/archaea. It would be important to explain here why only 136 families were included in the analysis — is it due to an a priori choice based on specific functions (i.e. only families linked to eukaryotic apomorphies), due to some sort of gene tree quality filtering, or both? If quality filtering was applied, how many families were initially surveyed, and which criteria were used to clean the list? Some of these details are provided in the supplementary notes but making them explicit in the main text (and referencing where to find more information) would be very useful at this point.

L120 "Our approach could not be applied to novel eukaryotic genes that were duplicated prior to LECA because, without an extant prokaryote outgroup, the ages of these duplication nodes would be highly sensitive to assumptions about the maximum root age." This appears to me as an important point but its interpretation is a bit unclear — is it possible to apply the method but the results would be unreliable; or is it completely impossible to do so? Given that gene trees are dated using a calibrated species tree as reference, why does the lack of outgroup affect the age of root of the gene tree, provided it's possible to identify it with confidence? Wouldn't it be impossible to use information from such eukaryotic-specific paralogs, if one cannot ascertain whether they are of bacterial or archaeal descent? Please clarify.

L150: what do the horizontal span of each non-alpha-proteobacterial gene in Fig1D represent? What do the various horizontal lines indicate? Are blue lines the "node average age"? This is highly unclear.

L160: I assume these gene families were among the initial 136 gene trees used in the analysis above?

L337: mitoFECA should be mFECA?

L377: could the authors provide an estimated date range for the GOE here, for ease of comparison?

L447: presumably the authors mean that debate has been "constrained" in general, rather than "poorly constrained"? Or is this a statement against unwarranted speculation on this issue?

L790 "but plasmid descent differs by secondary plasmid acquisition events, where nodes contained the same set of taxa they were cross-braced." — What does "cross-braced" mean in this context?

L460: This summary Methods section is poorly written (including punctuation), and it references quite a few terms and concepts that are not present elsewhere in the main text — e.g. "domain origination" or "domain analysis". The authors

should put these methods into a better context for the non-expert reader. Consider expanding this effort to the otherwise quite schematic Supplementary Note methods.

Comments on Figures

The general layout of Figs 1 to 4 is very informative as it nicely links together species- and gene-level phylogenies with the temporal dimension of this study. However, the figures contain many details, acronyms, and marks that are often left unexplained in the main text and legends (and describing main figures in supplementary notes is quite unusual), and panels are often referenced out of order (Fig3 and 4 come before most of Fig2). I list some of these problems below.

Beyond the main figures, the ordering of the Supplementary Figures and Notes does not match the point in which these materials are referenced in the main text, which unnecessarily complicates things.

Fig1E: it is not clear what the vertical blue/black lines mean in this context.

Fig2. The meaning of the green dots, squares and diamonds in the figure should be explained in the legend? What does "HDP" mean in this context? (presumably something related to the density distribution, but this should be made clear). There are no gene families indicated with " α -| " — do the authors mean "a" or "A", as this refers to asgard archaea? There are similar issues in Fig3.

Comments on the supplementary materials/notes

In Supplementary Notes 1 and 2 the methods are explained with an adequate level of detail, but in a very schematic way that complicates things unnecessarily. Some introductory remarks describing what is to be found in each section would be welcome. I acknowledge that this is provided in the main text summarised Methods section, but going back and forth between the two documents is an unnecessary hassle.

In Supplementary Note 3: "determine the impact of alternatives on our results." Be more explicit: alternative roots?

Sadly, some figures in the supplementary notes are not provided with sufficient resolution (e.g. species trees such as Supplementary Figure 7-9). This should be revised.

(Remarks on code availability)

The supplementary notes contain some inline code invocations, but they also reference some scripts and directory structures which I could not find in the SM or any associated data repository.

Version 1:

Reviewer comments:

Referee #1

(Remarks to the Author)

The authors have addressed all the key points raised in reviews.

The paper is much better for the revisions and constitutes a significant advance in the field.

It will change the way we see eukaryogenesis.

(Remarks on code availability)

Referee #2

(Remarks to the Author)

I co-reviewed this manuscript with one of the reviewers who provided the listed reports.

(Remarks on code availability)

Referee #3

(Remarks to the Author)

Having reviewed the revised documents and the response to reviewers document, I commend the authors for their hard work. I have no further comments on the manuscript.

(Remarks on code availability)

Referee #4

(Remarks to the Author)

The authors addressed all my questions regarding the data set and methods, and revised the text to take into account my feedback on the methodological limitations and the relatively limited comparison with previous studies. I particularly appreciate the clearer explanation of the data sources used in the analyses. I also greatly appreciate the expanded discussion, especially the section on the energetic requirements for the evolution of eukaryotic complexity and the broader perspective that emerges from the results of this work.

I was a bit surprised that TransDecoder was used for ORF prediction in genomes (as it is usually used for transcriptomes), but I assume that it can work quite well in the case of prokaryotic genomes as well.

I have no further comments and strongly believe that this is a very valuable paper to advance the discussion on the early evolution of eukaryotes, and hope that it will be published soon.

(Remarks on code availability)

Referee #5

(Remarks to the Author)

I commend the authors for their thorough consideration of the points raised in the previous round of revision, especially the figures and the (im)possibility to use eukaryotic-specific paralogs in their clock analyses. Both the methods and the figure legends are now much clearer. The updated discussion is learned, balanced, and informative.

(Remarks on code availability)

I have checked the inline code supplied in the supplementary nodes, but the Zenodo links provided above are private.

Peer review report for Kay et al. 2025-03-07178 "Dated gene duplications elucidate the evolutionary assembly of the eukaryotic cell"

Referee #1

Summary

In their study 'Dated gene duplications elucidate the evolutionary assembly of the eukaryotic cell', Kay and colleagues use a geological record-anchored molecular clock to date key gene duplication events that occurred during the evolution of eukaryotes. While time estimates remain wide, this allows the authors to identify genes that likely duplicated early on in eukaryotic evolution (prior to the acquisition of the alphaproteobacterial ancestor of the mitochondrion); those that duplicated around the time of mitochondrial domestication; and those that duplicated later along the path to LECA.

In analyzing their findings, the authors identify duplication events that are consistent with the emergence of new functions which are seen as instances in which functionally related machinery duplicates at a similar time. These events can be used to constrain hypotheses about the likely path of eukaryogenesis.

Amongst this set of gene duplications, there are several that are notable, including the following:

i) The authors identify duplications in genes with both archaeal and bacterial lineages involved in embedding mitochondrial endosymbiosis. These include: MRPL45 – a mitochondrial ribosomal protein, CARS2 – an archaeal derived tRNA synthetase operating in mitochondria (could this be related to Fe-S formation?), and the protein import system TIM14. Importantly, all appear after the "great oxygenation event."

ii) DNA damage repair system evolution is striking because of its mix of early archaeal proteins and late bacterial-derived proteins. This analysis has implications for the dating of the evolution of meiosis, and suggests a new way to think about DNA damage-repair in human disease.

In a field that has too many hypotheses and not enough data, this paper marks a very important step forwards. For this reason, the data presented will interest a very wide audience of evolutionary and molecular cell biologists and should spawn research into the evolution of many systems whose emergence via a gene duplication is touched on in the paper.

Overall, the breadth and depth of the analysis carried out is impressive, the paper is well written, the Figures are relatively clear, the discussion is excellent, and the text is supported by a long, but very helpful supplement.

There are a couple of issues that we suggest be addressed prior to publication.

Major comments:

(Comment 1) The analyses are based on the use of a limited sample of Asgard genomes. In doing so, they have excluded representatives of the Hodarchaeales. This seems like an unfortunate omission, since these Archaea have been suggested to be the closest archaeal relatives of eukaryotes. We would therefore strongly suggest that the authors consider including additional Asgard genomes in their analyses.

Thanks for pointing this out. We did actually include two Hodarchaeales MAGs in our analyses, but their identity was not clear in the text or SI. In the revision, we have (i) added a note to Supplementary Table 2 and indicated the Hodarchaeales, and (ii) updated the species names in Supplementary Table 2 to match the tip names of our species tree.

*“Highlighted in the table (by *) are two Hodarchaeales as defined by GTDB, but which are labelled by NCBI as Heimdallarchaeota.”*

Supplementary Table 2

We include here the position of the two Hodarchaeales () in a subset of the maximum likelihood tree used to build the species tree. In our analysis, the Hodarchaeales branch together, but we find eukaryotes to be sister to*

Heimdallarchaeia as a whole, which are further explored in the topology tests of Supplementary Notes 3.

In these tests we find setting eukaryotes as sister to Hodarchaeales did change the age of nFECA to slightly younger ages (-0.19 Ga). These results suggest that root placement amongst Heimdallarchaeia can produce small changes to nFECA age, but that the relative ages of our key nodes and our proposed timeline for eukaryogenesis remains robust.

“To investigate the effect of the placement of eukaryotes within the Asgard archaea, we set nFECA sister to Hodarchaeales³. This test produces a modest decrease in both nFECA and mFECA ages (2.92 to 2.65 Ga; 2.30 to 2.23 Ga), but the two FECA branches remain different lengths, with non overlapping CIs on their node age, in this analysis LECA remains unchanged.”

Supplementary notes 3

(Comment 2) The paper implies that there was a burst of gene duplication events in the nFECA lineage that was associated with the acquisition specific eukaryotic cell functions. There seems to have been no attempt to measure the change in rate, even though the paper states: “our analyses suggest that in the nuclear lineage the rate of duplicate gene fixation is even higher than in other Asgard.” More evidence to support this statement would be welcome.

Thanks for raising this interesting point. Quantifying the rate (e.g., duplications per million years) across Archaea would be interesting, but it is outside the scope of our analyses here. The reason is that we searched for gene families that show pre-LECA duplications, whereas comparing duplication rates between Asgard and other archaea would require a systematic analysis of archaeal gene families more generally (e.g., with a reconciliation model). We have now clarified the language in the main text, to avoid confusing the reader into thinking that we might be seeking to infer rates, since we are only considering two branches on the species tree.

“To investigate the timing of gene duplications relative to LECA, nFECA, and mFECA, we surveyed a broad sample of eukaryotic, archaeal and bacterial genomes for gene families of prokaryotic ancestry with duplications that occurred within either of the two eukaryote stem lineages.”

Results - Establishing the eukaryogenesis timeline

(Comment 3) It would be useful on this point to include a more systematic analysis of duplication events that occurred earlier in the archaeal stem lineage (in Asgard, Heimdall and Hod lineages) that led to the divergence of homologs that retain functional differences in eukaryotes that are already included as subjects of the analysis in the paper. Examples of these could include: Sm proteins, ESCRTs, AAA-ATPases. Such an analysis would provide important context for the results presented in the paper. Some of these duplication and divergence events were likely important for the evolution of compartments prior to nFECA. While some Asgard duplications (e.g. tubulins) did not make it into the eukaryotic lineage, they could be used to provide evidence of selective pressure for duplication events in some protein families within the Asgard archaea/eukaryotes.

This is an interesting line of enquiry that we intend to follow up in a future study, although duplication of genes in the asgard lineage have been noted in other studies before. However, it is tangential to the focus of our manuscript which concerns the nFECA and mFECA stem-lineages and testing hypotheses of eukaryogenesis which concern innovations on these branches in the species tree.

(Comment 4) It would also be useful for readers if the paper had a fuller discussion about how definitions of nFECA are likely to change as more Asgard lineages and other potential close eukaryotic relatives are identified over the coming years.

Agreed. In the revision, we have added an explanation of this point to the introduction, writing:

“Note that nFECA and mFECA refer to the points at which eukaryotes branched from extant, sampled archaea and bacteria, respectively, on the tree of life. Their identities and ages will therefore change as more closely related genomes on either the archaeal or bacterial side of the eukaryotic family tree become available”
Results - establishing the eukaryogenesis timeline

(Comment 5) While the Figures are great, they still need work. To help readers, Figures and/or their legends should have a key that explains everything shown – e.g. GREEN SHAPES. It is not enough that this is explained in the text.

Thank you for pointing this out. In line with your suggestion, we have now added a key to all figures to improve clarity.

(Comment 6) In addition, the names of all proteins in Figures and related text should be consistent and easy to find, e.g. TREX1 and CENP-A.

To address this issue, we have modified Supplementary Table 7 to include a column identifying proteins appearing in (which) figures.

(Comment 7) “Compartment/organelle function” calling is dubious and should be avoided where possible. Instead of saying “vesicle trafficking” or “nucleus” it is much better to explain what the protein does or to state where it is localized in modern eukaryotes, and/or which other proteins it interacts with. This is especially true when we are considering the emergence of eukaryotic cell organization, since protein functions might emerge before the cell structure in which they act in extant eukaryotes. As an example of this, protein glycosylation occurs at the surface of archaea, but begins in the ER and is augmented in the Golgi in eukaryotes, so it’s better to state this as a protein activity related to secretion, not to an organelle.

Thanks for these helpful remarks - we have made a number of edits to better reflect these nuances in the revised manuscript. With respect to glycosylation we agree that our original formulation may be confusing and have rephrased potentially misleading wording.

We have also reviewed and updated semantics elsewhere in the manuscript. In some instances we kept the mentioning of organelles, because the inception of protein functions provides at least a relative constraint on the timing of origin of the organelles with which they are associated.

Other examples:

(Comment 8) ESCRT-III proteins, like CHMP4C, are membrane remodeling cytoskeletal proteins. There is very limited evidence to suggest that they play an important conserved role in kinetochore function.

We have added a reference that supports the involvement of CHMP4C, and also acknowledge that ESCRT-III proteins are primarily membrane remodelling proteins in the manuscript text.

“While ESCRT-III proteins are primarily involved in membrane remodelling³⁸, and cell division³⁹, CHMP4C (duplication date 2.77-2.06 Ga) may additionally be characterised in associating with microtubules⁴⁰ and attachment to kinetochores⁴¹ linking microtubule organisation to cell division”

Results - The cytoskeleton

(Comment 9) Histones and SMC proteins regulate chromatin architecture whether or not there is a nucleus.

We agree this is an important distinction, especially considering the function of these proteins in prokaryotes. We have not included discussion of these proteins in the manuscript section on the nucleus, they are discussed in the SI, and we have sought to clarify our wording the SI. The role of condensins and proteins functionally analogous to them are found throughout the tree of life, however the emergence of functionally distinct cohesins is a eukaryotic novelty, and relate possibly to a change in genome organisation.

“Based on the function of characterised SMC complexes in prokaryotes¹⁷, the ancestral complex likely had condensin-like function (that is, condensing chromosomes). Sister chromatid cohesion in cohesins appears to be a eukaryotic innovation, and from the role of cohesin in spindle assembly, meiosis and homologous recombination repair, suggest an adaptation to eukaryotic genome architecture.”

Supplementary Discussion - SMC family proteins

(Comment 10) And NLS sequences likely predate nuclei in many proteins where they may function to localize proteins to nucleic acids.

We agree with your suggestion, and have modified our discussion of nuclear localisation signals in the manuscript accordingly.

“ - although we note the possibility that nuclear localization signals might have evolved prior to the nuclear compartment, for example to localise these proteins to nucleic acids”

Results - The nucleus

(Comment 11) The Exosome and Sm proteins are involved in RNA processing in both archaea and eukaryotes. There is also evidence for nucleolar-like activities in archaea (Sulfolobales) and perhaps a nucleolar-like structure. This is important to be clear about, since the emergence of splicing, something that is specific to eukaryotes and depends on the presence of a nucleus, needs to be thought about very carefully.

Following from above we have modified the manuscript and SI with respect to the discussion on splicing and the nucleolar and nuclear compartments.

“Efficient biogenesis of the modern LSM complex requires the nuclear localization of its transient RNA partner, U6 snRNA, and, more broadly, the spatial separation of transcription and translation with or without a nuclear compartment appears critical for the emergence of regulated splicing. The early timing we infer for the evolution of spliceosomal components therefore supports the emergence of complex eukaryotic splicing before the mitochondrial endosymbiosis.”

Results - The nucleus

(Comment 12) As part of this, Figures are easier to make sense of when proteins with related molecular functions are placed together as has been done for much of the paper. This could be fleshed out in a table or additional figures.

- i) AAA ATPases (Cdc48, PAN, ATAD1, FIGN/Spast, Vps4...)
- ii) Lipid metabolism
- iii) DNA repair
- iv) DNA polymerases
- v) Microtubule function (Tubulin, gamma Tubulin, Dynactin, HATs, HDACs).

Thanks - this is a valuable suggestion. To this end, we have revised Supplementary table 7. For each row either a reference protein, or named HMM for the domain search is given, as well as named orthologs and NCBI gene ids in any of the LECA nodes. From this information it would be possible to re-encode the data in the table by GO term for any particular function, or domain of interest.

(Comment 13) Note that this type of analysis is especially informative when eukaryotic activities involve a combination of bacterial- and archaeal-derived components, e.g. in SFigure

We agree that processes with mosaic ancestries are particularly relevant for understanding the steps in eukaryogenesis, in our analysis we were only able to investigate 41 bacterial gene families, and we have presented as many as we could in the context of processes with gene families of archaeal descent. Unrepresented bacterial gene families belong to metabolic pathways for which we have no other duplicated members, although it would be interesting in the future to look at pathway mosaicism independent of pre-LECA duplications.

(Comment 14) In this sense, its worth highlighting the timeline in the evolution of i) Histones with tails (archaeal-derived), ii) Methyl transferases (bacterial-derived), and iii) HDACs (archaeal-derived). What about HATs?

Thanks for these interesting thoughts. We agree the development of chromatin is a very interesting aspect of eukaryogenesis, particularly the involvement of components of both archaeal and bacterial ancestry. While we were not able to piece together as many parts as we would have liked to have it as one the defining apomorphies on eukaryogenesis, we have followed the suggestion and have added a section covering the development of chromatin in the SI.

“Histones likely had regulatory roles in the archaeal ancestor 15, but acquired new modes of regulatory control during eukaryogenesis, including the emergence of N-terminal tails capable of being post-translationally modified, laying the foundation for eukaryotic chromatin-based epigenetic regulation. In our analysis, we resolve three families of histone modifying enzymes that underwent pre-LECA gene duplications, two of alphaproteobacterial origin (the METTL family, dated to 2.24–1.85 Ga, and SBNO1/2, 2.07–1.81 Ga), and one of inferred Asgard archaeal origin (the HDAC family, 2.26–1.98 Ga). The timing of these duplications are contemporary with or follow mitochondrial endosymbiosis suggesting that the elaboration of eukaryotic chromatin regulation was a relatively late event in the eukaryogenesis timeline and involved gene families from the mitochondrial endosymbiont.”

Supplementary Discussion - Chromatin

(Comment 15) It would be good to include a fuller general discuss of how all the data taken together shed light on the relative timing of gene duplications associated with the acquisition of activities that are now confined to the ER, Golgi, nuclear envelope, plasma membrane, mitochondrial outer membrane, endosomes, and the lysosome (Rag GTPases / TOR complexes / V-ATPase etc).

The emergence of the following genes seems an important early marker of this: SGPP1/2 SRD5A1 2.86-2.26 Ga; DPM2/PIGP 2.89-2.08 Ga; STT3A/B 2.80-2.20 Ga).

Thanks - we have extended our description of the endomembrane section of the results clarifying that there was a late period of complexification of now ER associated functions relating to bacterial integration.

“Some later gene duplications in families of archaeal origin are consistent with integration into these pathways of bacterial origin. For example, SGPP1/2 is an enzyme of archaeal origin that today carries out an intermediate step in eukaryotic sphingolipid biosynthesis, and we infer its origin via gene duplication diverging from DOLPP1 in the archaeal dolichol glycosylation pathway around 2.67-1.98 Ga. Pre-LECA duplications in the other gene families of bacterial-origin involved in membrane lipid biogenesis also duplicated at around this time (Fig. 4B, GPAT3/4 and LPCAT1/2/4 2.54-2.06 Ga; SPTLC1-3 2.16-1.99 Ga; ACSL1/3-6 2.12-1.87 Ga). Gene trees for two of these families (ACSL, GPAT/LPCAT) do not provide compelling support for an alphaproteobacterial origin, with some (ACSL) consistent with acquisition from other bacterial groups such as Myxococcota 46,47. Together, these

duplications point to a phase of late complexification in ER-associated functions, reflecting metabolic integration of archaeal and bacterial membrane biosynthesis pathways. This finding is in agreement with hypotheses in which the membrane transition involved acquisition of bacterial genes from non-mitochondrial sources.”
Results - The endomembrane system

By contrast, our analyses did not provide a firm conclusion for archaeal duplications in the development of the lysosomal compartment (through the Ragulator and V ATPase complexes), although we appreciate the insight, so we have not elaborated on this in the text.

Minor comments

(Comment 16) It would be good to be clearer in the main text about the potential function and source of bacterial genes in 1E that duplicated early.

Our analysis revealed a limited number (2) of bacterial genes that duplicated before the mitochondrial endosymbiosis, making it difficult to confidently identify a specific enriched function. Both identified genes are involved in membrane biogenesis, and further investigation into their wider pathway could potentially provide a consistent signal of origin, perhaps within a syntrophy-compatible regime. However, our current data are insufficient to make a definitive statement, although we have discussed the context of these two families in the manuscript.

(Comment 17) The key for Figure 1B refers to mFECA as 22.0Ga.

Corrected.

(Comment 18) Authors refer to both ‘Heimdallarchaeota’ and ‘Asgardarchaeota’ in the text. Authors should adhere to one naming system.

Thank you, yes, we have used reference to Heimdallarchaeia, once, to be specific about placement in the species tree,

“Within the phylogenetic tree which was estimated under Maximum Likelihood, nFECA branches within Asgard archaea18 (also known as Promethearchaeota19), sister to Heimdallarchaeia, and mFECA is sister to Alphaproteobacteria to the exclusion of Magnetococcales, consistent with recent analyses 20–23.”
Results - establishing the eukaryogenesis timeline

but all other times have preferred to use Asgard archaea.

(Comment 19) In Figure 2A, it is odd to refer to cytoskeletal genes as “vesicle trafficking genes.” Just call them “cytoskeleton genes”.

Corrected.

(Comment 20) CHMP4C is an ESCRT-III homologue.

We expand on our correction of this above (Comment 8).

“While ESCRT-III proteins are primarily involved in membrane remodelling 38, and cell division 39, CHMP4C (duplication date 2.77-2.06 Ga) - ” manuscript

(Comment 21) Some author comments on the Supplemental text have not been addressed or deleted!

Corrected.

(Comment 22) METTL4, not “METL4”, belongs to a subclade of MT-A70.

Corrected.

(Comment 23) There is no evidence to show that cytoskeleton “pattern” protrusions in Asgard. All one can say for the moment is that they are present in Asgard cells and in their protrusions.

Thanks for this important clarification. We agree that current data do not conclusively demonstrate that cytoskeletal elements pattern the membrane protrusions observed in Asgard archaea. Our intent was to highlight the co-occurrence of actin-like proteins and cellular protrusions in Asgard as suggestive—but not definitive—of a functional link. We have revised the manuscript accordingly

“Cultivated representatives of extant Asgard archaea 14 have protrusions from the cell body, potentially as a means to engage in metabolic exchange with other cells 34, and specific homologs to eukaryotic cytoskeletal proteins have been identified and characterised 14,35.”

Results - The cytoskeleton

(Comment 24) It is not certain that phagocytosis requires branched actin. All it needs is binding, membrane spreading and scission. Scission is aided by actin and by Dynamin. Did the authors look at Dynamin duplications? Did the authors look at any actin regulators? Formins?

We did attempt to explore both, however, they belong to large gene families for which we could not infer a reliable pre-LECA duplication topology.

(Comment 25) Microtubules are not always localized around MTOCs. This is only really true of some animal cells. Like actin, microtubules need nucleators - gamma tubulins.

This was a helpful comment, and we have revised the respective text accordingly.

“Our inferred early divergence of gamma tubulin (2.84-2.41 Ga, involved with nucleating and orienting microtubules), delta and epsilon tubulins (2.75-2.25 Ga, 2.61-2.07 Ga, paralogs localised to centrioles), gives evidence for early

establishment of microtubule organisation, with the filament forming alpha and beta paralogs emerging later”
Results - The cytoskeleton

(Comment 26) The following sentence is confusing: “The duplication of RNA polymerase complexes into RNA pol I, II and III is also consistent with the development of the nuclear compartment >2.4 Ga.” Why does this follow?

We have corrected the wording in this section to clarify our inference, again we have tried to separate function from compartment as per Comment 7.

“The duplication of RNA polymerase complexes into RNA pol I, II and III is consistent with changes to genomic organisation >2.4 Ga. In modern eukaryotes, RNA polymerase I is localised to the nucleolus and transcribes ribosomal RNAs; RNA polymerase II transcribes mRNAs, and RNA polymerase III transcribes tRNAs, 5S rRNAs and U6 snRNA, while the ancestral archaeal RNA polymerase complex performed all transcription. The divergence between RNA pol I and the others provides a minimum age for the development of a RNA pol I transcribed nucleolar compartment.”

Results - The nucleus

Referee #2

I co-reviewed this manuscript with one of the reviewers who provided the listed reports.

Referee #3

Kay et al. have dated and studied duplicate gene families across prokaryotic and eukaryotic diversity to elucidate the relative timing of key events in eukaryogenesis – including the endosymbiotic origin of mitochondria. The result is a remarkable dataset that provides many insights into thorny issues that researchers in the field of eukaryotic evolution have been grappling with. Some of the conclusions are complementary to other papers, such as rejecting ‘mitochondria-early’ scenarios in favour of increased complexity of ‘eukaryotic’ molecular and cellular features in archaea. But I am aware of no study that has addressed so many different elements of eukaryogenesis in such a synthetic and insightful manner.

I can’t find fault with the timescale / relaxed clock approach the authors have taken. It is far from perfect, but the approach has already been established, is clearly described, and the caveats and limitations are adequately addressed. There are no doubt conflicting signals and biases in the underlying data that push specific results this way and that; but the strength of the paper is in the sheer volume of data analyzed such that larger trends can be inferred. It really is a remarkable study that will serve the field well.

It’s not a perfect paper. There are errors and deficiencies throughout (see below), and while the main text is manageable in isolation, the volume of information presented in the

Supplementary Discussion / Notes is simply overwhelming. In some cases, it also seems redundant. See below for specific examples. I would encourage the authors to consider how much of the information in Supp info is truly necessary, and trim down accordingly.

Specific comments.

(Comment 27) Line 444: “CALM: Complexified Archaeon, Late Mitochondrion models of eukaryogenesis”. This is mentioned only once. I’m not sure the field would benefit from another acronym, in this case one that refers to a certain flavor of eukaryogenesis models. I don’t feel strongly about it, but I doubt it would be very useful.

Debate over eukaryogenesis has a number of acronyms because there are a number of competing hypotheses for eukaryogenesis and the acronyms are used in shorthand to discuss these hypotheses. Our results are incompatible in detail with all existing hypotheses of eukaryogenesis and, as such suggest a distinct scenario. We have proposed an acronym for this scenario to aid future debate in which it will doubtless be weighed against the existing plausibility-based hypotheses.

(Comment 28) Redundant main text and supp info figures. Figure 2 in the main text (endomembrane system) has a counterpart: Supp Figure 15. They are different, but only slightly so. I can’t tell the difference between Figure 4 and Supp. Figures 17. Why not replace the former with the latter? It’s not clear to me why there is so much redundancy between main text figs 2-4 and the Supp figures.

Thanks - these are useful points. We agree that there is a lot of redundancy in the supplementary figures, and we have removed 4 supplementary figures, Fig 2 is expanded across supplementary figures 9 and 10, to better place these genes within a nuclear context.

(Comment 29) Figures in the Supplementary Notes – a good number of them cannot be read in the pdf file that I reviewed (e.g., Fig S7, S9). If they can’t be read when zoomed in, then there is no point in having them be a Supp figure).

We agree, the dendrograms are too large to be included in the SI, and we have removed them accordingly. The same information is provided as Newick-formatted tree files, provided in the supplementary data.

(Comment 30) Line 93/94. Redundant with statement and abbreviations in the introduction (lines 67-69)

Thanks for spotting this - we have now removed the redundancy.

(Comment 31) LBCA used only twice in MS, on lines 104 and 107. Spell out for non-specialists.

We have altered the manuscript text for readability as suggested, but introduced and retained the acronym in the first instance for its appearance in the figures.

(Comment 32) Line 119 – ‘total-groups’ = typo? If not, didn’t make sense to me in this context.

We have modified the text for clarity.

“The age of these duplications relative to LECA, nFECA and mFECA constitute the focus of our study. This was achieved by dating paralogue divergences within constraints on the age of LECA and its archaeal- and alphaproteobacterial-derived total-groups, established in our earlier dating of these species divergences.”
Results - Establishing the eukaryogenesis timeline

(Comment 33) Line 132: “Based on these considerations, we hypothesize that the initial accumulation of gene duplications in the mitochondrial lineage, after its divergence from Alphaproteobacteria, must reflect a shift to a more eukaryote-like population-genetic environment.” Pop gen factors certainly would have played a role in the likelihood of duplicates being fixed, but another factor is that proto-mitochondrial duplicates would have been exploring new biological / biochemical landscapes in the context of a proto-eukaryotic cell. Things like TIMs are mentioned, of many possible functions that would have been massively rejiggered during this tumultuous time of cell evolution, not to mention alphaproteobacterial-derived genes / proteins that ended up having nothing to do with the mitochondrion. I guess what I am saying is that there is fixation and then there is functional divergence – which of course are related – but it’s possible for dups to be fixed with little / no functional divergence and this paper is all about eukaryotic diversification, not population genetics. So I suggest perhaps rethinking the language a bit at this point of the text to go more along with the main focus.

Thanks, we agree with this point. As noted by the reviewer, the functions of some of these alphaproteobacterial duplicates in the wider eukaryotic cell provides the most direct evidence that mitochondrial endosymbiosis had already begun at this time. We have revised the text accordingly to focus more directly on this point.

“Among the earliest alphaproteobacterial-origin duplications are gene families that gave rise to paralogues which today function outside the mitochondrion (e.g. exclusively nuclear functioning RNA methylase METTL4, 2.24-2.11 Ga; type Y family polymerases, earliest duplication 2.24-2.12 Ga, and mismatch repair proteins, earliest duplication 2.18-2.08 Ga), we suggest that the initial accumulation of gene duplications in the mitochondrial lineage evidences the onset, or establishment, of mitochondrial integration into the eukaryotic cell. In further support of this we find alphaproteobacterial origin genes coding for proteins involved in the functioning of mitochondrial endosymbiosis (preprotein import system TIM14 and TIM44 gene families, 2.24-1.99 Ga and 2.22-2.00 Ga, respectively) with similarly early duplication dates.”
Results - Dating mitochondrial endosymbiosis

(Comment 34) Figure 1 and legend needs some work.

We have addressed this point (see below).

(Comment 35) plastid, not plasmid

Corrected.

(Comment 36) Part D is not visually distinct from parts C and E (maybe lower the 'D' so it is less seemingly part of the bottom of the histogram in C).

Agreed, we have revised the layout of Fig. 1.

(Comment 37) Part E. What are the horizontal lines, exactly? They are the 19 non-alpha dups, right? Can they be labeled individually, or at least explicitly labeled as a set so that they at least can be seen as 'other bacterial duplicates? Why do some have 1 light blue ticks at right end, but some have a second internal blue tick? None of this is explained in the legend at all.

Thanks, we have corrected this oversight by revising Fig. 1 to indicate the meaning of these lines. What we intended to do was show the date of divergence of a bacterial gene family from other bacteria (black bar) and pre-LECA duplications (blue bar), the horizontal line just provides a way of linking these pieces of information, we hope that the changes to the figure now make this clear.

(Comment 38) Finally, I'm not sure this figure needs the labels at the very bottom (nFECA, mFECA, etc.) to be labeled part F. They align with the icons and dates at the top of the right side of Figure 1 (i.e., right above the histogram in part C), but the images at the top don't have their own label.

Agreed - we have now removed the unnecessary labels.

To conclude: while the data in Figure 1 right are very interesting and informative, it takes a lot of staring to figure out what exactly is being shown. I encourage the authors to re-think the layout a bit.

Figure 2.

(Comment 39) If I am interpreting the data correctly, delta and epsilon tubulins seem to be older than the alpha/beta paralogs. Brief comment on the cell biological significance of this WRT microtubule evolution for centrioles, etc.

Thanks for this observation, we have updated the text for this section to better explain the role of the tubulin paralogs, and possible development of the tubulin system.

"Our inferred early divergence of gamma tubulin (2.84-2.41 Ga, involved with nucleating and orienting microtubules), delta and epsilon tubulins (2.75-2.25 Ga, 2.61-2.07 Ga, paralogs localised to centrioles), gives evidence for early establishment of microtubule organisation, with the filament forming alpha and beta paralogs emerging later"

Results - The cytoskeleton

(Comment 40) Error in top right. The main part of the microtubule is a alpha-beta heterodimer, but the figure includes gamma in the dimer. This error also appears in Supp Figure S16.

Corrected.

(Comment 41) gene families of Asgard archaeal origin are shown with an 'a' in the figure but described with an 'alpha' symbol in the legend. This 'a' / 'alpha' issue applies to Figures 3 and 4 as well.

Fixed by creating a new figure key.

(Comment 42) General comment – the 'a' / 'alpha' paralogs have clear asgard homologs and thus are special, more special than is obvious from the figure. Beyond sorting out the 'a' / 'alpha' issue, I would suggest making them stand out more in the figure. Perhaps with a colored circle with an 'a' / 'alpha' symbol inside it? There is already a ton of data in these figures, and the tiny 'a' / 'alpha' beside the paralog names almost goes unnoticed.

This is a good suggestion, we have altered the visual format to make the origin more pronounced, and fully described in the figure as a key.

Figure 4.

(Comment 43) What does 'Genes duplicated into the mitochondrion' mean (top left)?

Apologies; we meant "mitochondrial proteins of archaeal descent", which we have now used in the figure text.

(Comment 44) The text refers to 4C, but there is no such part C labeled in the figure (it's obviously the bottom part analyzing the CLCN paralogs involved in endosome / lysosome functions).

Thanks, Corrected.

(Comment 45) Line 158: "While many of the genes that specify these structures are specific to eukaryotes, others descend from prokaryotes." It's a nit-picky point, but technically ALL the genes that specify "these structures" descend from prokaryotes, with the exception of cases of novo gene evolution from non-coding DNA. De novo gene evolution is now recognized as 'a thing', but I think a minor rephrase is needed to avoid confusion for non-specialists. Many and probably most 'eukaryote-specific' genes ultimately came from prokaryotic homologs but are no longer recognizable as such. 25 / 30 years ago we didn't realize that prok. Homologs of tubulins, actins etc. can in fact be found, with the help of 3D structures, despite the extreme primary sequence divergence. Indeed, they are discussed in the very next paragraph.

This is an interesting discussion, and important to point out. We have revised the text accordingly, writing:

“While many of the genes that specify these structures are or appear to be specific to eukaryotes, others have clear recognisable descent from prokaryotes.”

Results - The development of key eukaryotic traits

(Comment 46) Line 215 – ER-localized proteins, not ER-localized genes

Corrected.

(Comment 47) Supp Info files and Supp Notes are out of order. The first SI file mentioned at the very beginning of results is Fig S10, the first Supp Note is S3.

Thank you for pointing this out: we have made sure that all Figures and Supplementary Figures appear in correct order as referenced in the main text., Supp Note 3 is still an exception, we defend this as its content is logically dependent on Notes 1 and 2.

(Comment 48) Line 300 – “We identify 4 gene family paralogs of archaeal descent, whose other gene family members occupy different compartments (ATAD1 2.63-2.13 Ga; TOP3A 2.6-2.2 Ga; IFIH1 2.37- 1.84 Ga; CARS1 2.14-1.87 Ga; Fig. 3A, Supplementary Discussion 1) in the interval 2.63-1.87 Ga.” You mean Figure 4, right?

Corrected.

(Comment 49) Discussion – GOE is described very late in the MS and not actually spelled out. This could be confusing to non-specialists, since based on context alone it could be assumed to be great oxygenation or great oxidation. So I would spell it out and have a look at this section of the text so that the main message to readers is clear and unambiguous.

Thanks - we have now spelled it out, and elaborated a little more on it in the Discussion.

“There has been considerable debate about the environmental context of eukaryogenesis and of eukaryote diversification. It has long been argued that oxygenation of the biosphere, the Great Oxidation Event (GOE, 2.43-2.22 Ga⁸⁴), was an environmental driver underpinning mitochondrial endosymbiosis and the origin of eukaryotes (e.g. Sagan 1967 85) and, indeed, our evolutionary timeline precludes a syntrophic association between nFECA and mFECA prior to the GOE. However, the formative stages of eukaryogenesis likely took place under anoxia or hypoxia since oxic conditions were limited to the surface waters of the Earth’s oceans for much of the Proterozoic 86,87. Specifically, our analyses point to an origin among archaea in the late Archaean, mitochondrial endosymbiosis almost coincident with the GOE, and diversification of crown-eukaryotes in the late Palaeoproterozoic.”

Discussion - Timing eukaryogenesis

Referee #4

A. Summary of the Key Results

The authors address a long-standing and fundamental question in evolutionary biology: the timing of key events during eukaryogenesis. This is a particularly difficult question because there are no intermediate stages and few fossils. Various approaches have been developed to answer this question, often focussing on gene duplication events. Here, the authors use a relaxed molecular clock approach applied to gene duplications and reconstruct a timeline that suggests that key cellular features (cytoskeleton, endomembrane system and nucleus) evolved prior to mitochondrial endosymbiosis. They propose that nFECA evolved between 3.1–2.8 Ga, mFECA between 2.4–2.2 Ga and LECA around 1.8–1.7 Ga.

B. Originality and Significance

This study is highly relevant and methodologically advanced. While the biological conclusions are consistent with previous work (e.g. Pittis & Gabaldón 2020; Vosseberg et al. 2021) already suggested substantial host complexity prior to mitochondrial acquisition, the authors reinforce these ideas with newly compiled datasets and careful dating of the evolutionary events. Explicitly highlighting the methodological novelties, such as specific improvements to the datasets or refinements to the analyses, would further enhance the originality of the study.

Overall, the study makes an important contribution by rigorously testing an important evolutionary hypothesis with an advanced methodology and a large dataset.

C. Data & Methodology: Validity of Approach, Quality of Data, Quality of Presentation

The methodology is well chosen, carefully explained and replicable. The selection of hallmark genes associated with important cellular systems provides a solid biological basis for the evolutionary interpretations. However, some clarifications are needed:

(Comment 50) It would be helpful to make clear how the dataset differs from previous studies, in particular, the selection criteria for the proteomes used and the genes used in the analyses.

Thanks, we have specified more clearly our taxa and gene choice at the beginning of the Results.

“Our phylogenetic analyses were based on a concatenation of 64 marker genes including taxa chosen to provide a phylogenetically representative sample of eukaryotes, bacteria and archaea, particularly including the closest relatives of eukaryotes and taxa for which geological calibrations are available.”

Results - Establishing the eukaryogenesis timeline

(Comment 51) The term "downloaded proteomes" should be defined more precisely. Are they predicted proteins from complete genomes? Were all available proteomes used or was filtering performed?

Thanks - this is an important clarification. We have consequently updated the manuscript methods and the SI. For the majority of the species, proteomes were downloaded from either EukProt or the NCBI database. For a small number of prokaryotes, proteomes were predicted from genomes using TransDecoder. We

have clarified these by adding a column to Supplementary Table 2 and by revising the corresponding paragraph in the Methods section.

“Species data were obtained either from EukProt 104 or from NCBI. We relied on gene predictions when present and otherwise, predicted open reading frames using TransDecoder, see Supplementary Notes 1, Supplementary Table 2.” Methods

*“With the exception of a few prokaryote species, we used publicly available predicted protein sets from either NCBI or Eukprot. When unavailable, we predicted open reading frames from publicly accessible genomes using TransDecoder.LongOrfs. In addition these proteomes were further processed by removing sequences of less than 50aa (unlikely to be useful in our analysis) and deduplication of 100% identical sequences, using cd-hit (-c 1.0). The data source, and whether proteomes were predicted are indicated for every taxon in Supplementary Table 2.”
Supplementary notes 1 - Source data*

The figures are very nice and informative and help a lot to convey the results.

D. Appropriate Use of Statistics and Treatment of Uncertainties

(Comment 52) The statistic is appropriate and the methodology is applied with care. Nevertheless, further discussion of the assumptions and known limitations of molecular clock models, especially with respect to deep-time phylogenies, would be valuable. In particular, genes acquired through endosymbiosis or HGT might have different duplication rates compared to native archaeal genes, which could affect clock estimates even in a relaxed model. Addressing this point would increase confidence in the results.

We have clarified some of the limitations of clock models in our Discussion section ‘Comparing clocks to other methods’

“Second, clock models provide a more flexible way to model variation in evolutionary rate through time, based on all of the available calibrations and sequence data - though we acknowledge that the real pattern of rate heterogeneity during eukaryogenesis and its associated horizontal gene transfer and duplications is likely to be more complex than captured by any current methodology.”

In addition to outlining the strengths of the clock approach compared to previous methods, we also highlight some limitations, writing:

“We note that this hierarchical framework implies that the inferred ages of gene duplications are informed by the ages of nodes on the dated species tree. For example, the inference that mFECA is younger than nFECA (Fig 1. 52,53) supports younger ages among duplications of alphaproteobacterial than Asgard origin, although sensitivity analyses demonstrate that this conclusion is robust to substantial variation in species tree ages”

Supplementary Notes 3

E. Conclusions: Robustness, Validity, Reliability

The conclusions are robust and generally well supported. However, some points could be expanded:

(Comment 53) Are there specific functional categories associated with duplications after mitochondrial acquisition?

As we expand on below, very few archaeal gene families begin to duplicate after mitochondrial acquisition, but many more begin duplication before and continue to duplicate up to LECA. By proportion of families undergoing their first duplication, those of bacterial origin dominate in the periods after mitochondrial acquisition. They are associated with three functional processes in the main, the establishment of the mitochondrion (discussed in the manuscript), the elaboration of nuclear processes including DNA repair and meiosis (Supplementary discussion), and duplication of genes involved in isolated specific metabolic processes (some of these are orphaned from the main discussion but they are tabulated in Supplementary Table 7).

(Comment 54) Are there recognisable patterns by which archaeal genes duplicated before versus after mitochondrial acquisition?

With respect to genes of archaeal descent, only a few archaeal gene families (5 of the 95 in our set) undergo their first duplication after mitochondrial acquisition. One of these is to support mitochondrial protein synthesis (CARS1/2), the remainder have nuclear functions which is perhaps unsurprising as many of our archaeal origin gene families are nuclear.

(Comment 55) The energetic cost of complexification remains a major argument against a mitochondrial late scenario; this should be discussed further.

We agree that energetic constraints have shaped hypotheses about the timing of mitochondrial acquisition. However, we note that this argument is largely based on theoretical models of bioenergetic potential and plausibility rather than direct phylogenetic evidence. While our findings do not directly address energetic feasibility, they suggest that mitochondrial acquisition (and an implied metabolic capacity), was not a strict precondition for the initiation of eukaryotic cellular complexity. We have now expanded on this point in the revision.

“Setting this age for we find that 85% of the pre-LECA duplicates of archaeal origin were estimated to have been fixed prior to mitochondrial endosymbiosis, indicating that duplication was prevalent on the nFECA branch, and fixation of duplicate genes was not dependent on a change of metabolic circumstance associated with the mitochondrial endosymbiosis.”

Results - dating mitochondrial endosymbiosis

“While it has been argued that the alphaproteobacterial endosymbiont may have been an energetic requirement for the evolution of eukaryotic complexity 89, our results are instead consistent with views in which the early development of eukaryotic cell complexity was not dependent upon a mitochondrial partner, in agreement with recent theoretical work 102.”

Discussion - Testing hypotheses of eukaryogenesis

F. Suggested Improvements: Experiments, Data for Possible Revision

(Comment 56) There is no major flaw that would require new experiments. However, a clearer description of a selection of data sets, a slightly expanding the discussion of uncertainties, and better highlighting of methodological innovations would improve the manuscript.

We have modified the manuscript to include a clearer description of how and why the datasets were selected, as well as the uncertainties associated with our results. There are no specific methodological innovations employed in this study.

G. References: Appropriate Credit to Previous Work

The authors make appropriate reference to relevant previous studies.

H. Clarity and Context: Lucidity of Abstract/Summary, Appropriateness of Abstract, Introduction, and Conclusions

The manuscript is generally clear, logically organised and well written. A few minor points: (Comment 57) The novelty of the results is somewhat overstated in the abstract.

We have edited the abstract accordingly.

(Comment 58) The differences to previous work should be emphasised more clearly.

We have edited the manuscript to provide greater emphasis on previous work.

(Comment 59) There is a small inconsistency in the dating results between sections (e.g. 3.1–2.8 Ga vs. 3.05–2.79 Ga for nFECA); consistency should be ensured.

Thanks, we have corrected this oversight, for our time resolved data we have stuck to 2DP throughout, but for the development of features in general we have stuck to 1DP as we don't feel we can be more precise about the development of structures such as the nucleus.

Conclusion

This manuscript presents a compelling and carefully conducted study that makes a valuable contribution to the understanding of the evolutionary origins of eukaryotic complexity. To further enhance its impact, the authors should more clearly emphasise the methodological advances, expand the discussion of uncertainties and clarify some minor methodological details.

We thank the referee. We anticipate that we have addressed these constructively critical comments through the specific responses outlined above.

Referee #5

I have read this manuscript with great pleasure and interest. The authors set out to establish a chronology of the various events along the eukaryogenesis process by using dated gene

trees. They succeed in calculating reliable node ages for various gene duplication events functionally linked to various eukaryogenesis-related events, and use this information to provide an absolute and relative chronology of the process. One of the most interesting aspects of the study is that the authors go well beyond aggregating node age distributions, examining the implications for each gene family in quite detail. In addition, the use of a sequential Bayesian approach to time-calibrate single gene trees using a reference species tree is (to my knowledge) a welcome and important innovation in the study of early eukaryotic evolution. Even if it is used mostly as a point of reference for gene-level analyses, the construction of a robustly-dated species tree that reflects the dual bacterial/archaeal ancestry of present-day eukaryotic genomes is, in itself, a major achievement. The authors take great care in contrasting their findings with an array of previous hypotheses about the eukaryogenesis process, listing strong and weak compatibilities/incongruences with each of them in the Discussion section. This well-warranted caution does not preclude the authors from enunciating the fundamental characteristics of their eukaryogenesis model (early complexification, non-early mitochondrial acquisition...), and putting forward an accompanying acronym ("CALM"). The fact that the gene-level evolutionary histories supporting their results are readily available in well-annotated supplementary notes will greatly facilitate their examination by other researchers and prompt further research in this field.

Yet, there are some parts of the manuscript that could have been prepared with more care. For example, figures are often referenced out of order (especially the abundant supplementary materials) and incompletely described (with explanations being scattered between in-figure legends, captions, main text, and supplementary notes), and the summary Methods in the main text, while welcome, are hastily written (e.g. referencing concepts that do not appear elsewhere in the text). I list some of these issues in the line-by-line comments below.

Other than that, the phylogenetic analyses here presented are of very high quality, the results ("The development of key eukaryotic traits", L156-304) are described clearly, and the discussion derived from them is rich and well-supported (L305-onwards).

Comments on the manuscript

(Comment 60) L95: A brief description of how the 64 marker genes used in the species phylogeny were chosen would be most welcome at this point. Presumably, these include a mixture of genes whose eukaryotic homologs were inherited from archaea (nLECA branches) or bacteria (mLECA/pLECA branches), but that are otherwise shared among the various prokaryotic lineages, is that correct? The authors could also provide an indication of how many markers (and alignment positions?) fall within each category.

Thanks for this useful suggestion; we have made several changes in response: (i) we have rearranged the methods section concerning the species tree, we list the number of genes falling into those with LUCA, LBCA, and LACA nodes in their species tree, and given an approximate contribution of this towards the species tree concatenates.

"Of these, 25 had gene tree nodes corresponding to the last universal common ancestor (with a combined length of 6216 aligned amino acids), 40 with the last archaeal common ancestor (9283 positions), and 43 with the last bacterial common

ancestor (11716 positions). From this set we produced concatenates using a custom script (Supplementary Notes 2) for the whole time-resolved species tree (15761 positions) as well as subset specific concatenate alignments to explore prokaryotic and eukaryotic root placement within Archaea and Bacteria.”

Methods - Species tree topology

(ii) We have updated Supplementary Table 3 to give a gene by gene evaluation, again with alignment positions, the presence of key nodes, and any human orthologs.

“This table summaries the gene families used to produce species tree concatenates. For each gene family, the presence of key nodes in the single gene tree is indicated, as well as the trimmed alignment length (in the combined species tree concatenate). Additional columns identify the directory for its identification in the supplementary materials, as well as the presence of any human orthologs in the gene family.”

Supplementary Table 3

(Comment 61) L98: The sensitivity analyses mentioned here are an important part of the phylogenetic analyses of the study. The rationale behind their importance and the basic insights derived from them should be summarised here. Also, the authors should be more specific about where to find this in the supplementary Methods (here and elsewhere in the paper). In this instance, the term “sensitivity analysis” does not appear in the main text methods.

Thanks, we have done the following

(i) added a heading in the Methods outlining these tests.

“We extended our topology tests to investigate the impact of species tree topology and calibration on time resolution. These tests were focused on the impact in two main areas, the age of deep nodes in the tree (LACA, LBCA, LUCA), and the robustness of the eukaryogenesis timeline (m/nFECA, LECA), the results of these tests is given in Supplementary Notes 3, Supplementary Tables 4,5. In general we found the ages of the FECA ages, and their relative branch lengths to be robust to the uncertainty of deeper nodes in the tree (LACA, LBCA), changes to eukaryote species tree topology did slightly impact LECA ages but not FECA ages, or relative FECA stem length.”

Methods - Sensitivity Analyses

(ii) provided more explanations regarding this in the SI (Supplementary Notes 3)

(iii) explained the rationale before presenting respective results.

“We performed a range of sensitivity analyses to investigate the robustness of this timeline to alternative tree topologies, relative ages of key deeper nodes (such as LBCA and LACA), and the use of particular fossil calibrations (see Methods, Supplementary Notes 3); the results indicated that the age of the key nodes for the eukaryogenesis timeline (nFECA, mFECA and LECA, and in particular the time spans between them) were robust to the range of conditions tested (see Methods, Supplementary Note 3).”

Results - Establishing the eukaryogenesis timeline

(Comment 62) L104: “To investigate the impact of this uncertainty in LBCA age...” — “LBCA” means last bacterial common ancestor, correct? The rationale for these additional sensitivity analyses is not clear in the main text, not the least because the phrase “this uncertainty” implies that this issue has already been touched upon. Why is this problem not relevant in the archaeal/nLECA portion of the tree? Is it related to the relative age of the LBCA/LACA nodes? Is it due to differences in the amount of alignment positions (or number of reliable markers) in either of these areas of the tree?

Thank you, these are informative and instructive questions to ask. The reason for the discussion of the impact of LBCA age in the first version of the manuscript was that, when comparing the results of recent clock studies, ages for LBCA have been more variable than other key nodes. We therefore sought to determine the extent to which alternative reasonable ages for LBCA might impact the age of the key nodes for our analyses (nFECA and mFECA), and thus duplication ages. Our sensitivity analyses suggested that variation in LBCA age did not greatly impact the ages of these more recent nodes. We have now rewritten this passage to clarify this and explain the rationale for the sensitivity analyses.

(Comment 63) L116: The number of gene trees used in this study (136) is only a subset of all the eukaryotic gene families inherited from either bacteria/archaea. It would be important to explain here why only 136 families were included in the analysis — is it due to an a priori choice based on specific functions (i.e. only families linked to eukaryotic apomorphies), due to some sort of gene tree quality filtering, or both? If quality filtering was applied, how many families were initially surveyed, and which criteria were used to clean the list? Some of these details are provided in the supplementary notes but making them explicit in the main text (and referencing where to find more information) would be very useful at this point.

Thanks for this comment. In the revised Methods section, we now explain more clearly how we arrived at the final set of pre-LECA duplicates. Briefly, the final set was arrived at by an iterative process of quality filtering from an initial set of ~5000 HMMs “based on their widespread occurrence in eukaryotes, and functionally annotated representatives” (Methods - Source data and processing), identification of duplicate nodes was done by combining both automated and manual approaches. The final set of 136 gene families includes only those for which we could infer reliable gene trees, and for which a prokaryotic outgroup was available (discussed further in response to the next point below). Each gene family in our analysis is associated with two reports, the first detailing the identification of duplication, the second its time resolution, both reports present colour coded dendrograms from which we determined the gene family topology. These can be found in the supplementary data ‘Results’ section.

(Comment 64) L120 “Our approach could not be applied to novel eukaryotic genes that were duplicated prior to LECA because, without an extant prokaryote outgroup, the ages of these duplication nodes would be highly sensitive to assumptions about the maximum root age.” This appears to me as an important point but its interpretation is a bit unclear — is it possible to apply the method but the results would be unreliable; or is it completely impossible to do so?

Given that gene trees are dated using a calibrated species tree as reference, why does the lack of outgroup affect the age of root of the gene tree, provided it's possible to identify it with confidence? Wouldn't it be impossible to use information from such eukaryotic-specific paralogs, if one cannot ascertain whether they are of bacterial or archaeal descent? Please clarify.

Thanks for raising this important point. We were not as clear as we could have been about the rationale here in the previous version of the manuscript. During our initial investigation we identified many pre-LECA duplications of “eukaryote-specific” genes, i.e. apparently novel genes without a (detectable) prokaryotic outgroup. For these genes - unlike those of identified prokaryotic origin - the duplication node corresponds to the root of the gene tree. In initial analyses, we observed that when analysing these genes with our method, root ages consistently shifted towards the imposed upper limit constraint - consistent with the difficulties of estimating root ages reported in several previous studies for “deep time” molecular clock analyses (e.g. Moody et al. 2022, 2024; Davin et al. 2025). We conclude that the time resolution step in our method (specifically MCMCTree), requires some rate information from either side of a node to effectively place it, and consequently is less reliable for dating the root node than for estimating the age of later nodes in the tree. We have clarified this passage accordingly:

“In addition to duplication and functional divergence of bacterial and archaeal genes, eukaryogenesis also involved the duplication of genes novel to eukaryotes that originated prior to LECA. These genes were not integrated into our analyses because, within the gene trees, the duplication node is also the root and, therefore, sensitive to the root prior used in the clock analysis²⁴.”

Results - Establishing the eukaryogenesis timeline

(Comment 65) L150: what do the horizontal span of each non-alpha-proteobacterial gene in Fig1D represent? What do the various horizontal lines indicate? Are blue lines the “node average age”? This is highly unclear.

We have updated Figure 1D in response to this and other raised points to clarify this. We wanted to show in this figure the duplication history of non-alpha-proteobacterial gene families. The black vertical bars represent the node age of their divergence from other bacteria in our species tree, while the blue vertical bars indicate the ages of pre-LECA duplications. A black horizontal rule connects these nodes for each gene family, clarifying the link between divergence and duplication nodes.

(Comment 66) L160: I assume these gene families were among the initial 136 gene trees used in the analysis above?

Correct. To clarify this point, we have modified the manuscript text as follows:

“Among the 41 investigated gene families of bacterial descent, 19 did not provide

strong evidence of alphaproteobacterial origin. Similarly, 9 of the 94 archaeal gene families did not support an Asgard archaea origin.”
Results - Dating mitochondrial endosymbiosis

(Comment 67) L337: mitoFECA should be mFECA?

Thanks, corrected.

(Comment 68) L377: could the authors provide an estimated date range for the GOE here, for ease of comparison?

Thanks. We now introduce the GOE and give an estimate of its date range earlier in the Discussion.

“There has been considerable debate about the environmental context of eukaryogenesis and of eukaryote diversification. It has long been argued that oxygenation of the biosphere, the Great Oxidation Event (GOE, 2.43-2.22 Ga⁸⁴), was an environmental driver underpinning mitochondrial endosymbiosis and the origin of eukaryotes (e.g. Sagan 1967 85) and, indeed, our evolutionary timeline precludes a syntrophic association between nFECA and mFECA prior to the GOE. However, the formative stages of eukaryogenesis likely took place under anoxia or hypoxia since oxic conditions were limited to the surface waters of the Earth’s oceans for much of the Proterozoic 86,87. Specifically, our analyses point to an origin among archaea in the late Archaean, mitochondrial endosymbiosis almost coincident with the GOE, and diversification of crown-eukaryotes in the late Palaeoproterozoic.”
Discussion - Timing eukaryogenesis

(Comment 69) L447: presumably the authors mean that debate has been “constrained” in general, rather than “poorly constrained”? Or is this a statement against unwarranted speculation on this issue?

We have revised our wording accordingly.

“Debate over the process of eukaryogenesis has been constrained because there are no extant lineages representative of the component steps in building a eukaryotic cell.”

Conclusions

(Comment 70) L790 “but plasmid descent differs by secondary plasmid acquisition events, where nodes contained the same set of taxa they were cross-braced.” — What does “cross-braced” mean in this context?

We have made a small expansion on this point in the Figure legend:

‘Where nodes containing the same species are repeated on the species tree, the nodes are cross braced (as 43) fixing them to the same age’.

(Comment 71) L460: This summary Methods section is poorly written (including punctuation), and it references quite a few terms and concepts that are not present elsewhere in the main text — e.g. “domain origination” or “domain analysis”. The authors should put these methods into a better context for the non-expert reader. Consider expanding this effort to the otherwise quite schematic Supplementary Note methods.

We have extensively revised the SI and the Methods accordingly. With regards to these pipeline specific points, we have summarised both pipelines succinctly in the Supplementary notes 1 focusing on their methodology, whilst moving the surplus material concerned with the running of these pipelines to readme files in the supplementary material.

Comments on Figures

(Comment 72) The general layout of Figs 1 to 4 is very informative as it nicely links together species- and gene-level phylogenies with the temporal dimension of this study. However, the figures contain many details, acronyms, and marks that are often left unexplained in the main text and legends (and describing main figures in supplementary notes is quite unusual), and panels are often referenced out of order (Fig3 and 4 come before most of Fig2).

We hope to have addressed these with some other general changes to the figures, as well as by removing some of the redundancy by;

- (i) Removing 4 redundant supplementary figures.*
- (ii) Re-ordering the supplementary figures such that their occurrence in the manuscript is in the correct order.*
- (iii) Adding a figure key to all figures which present gene duplications.*
- (iv) Additional text within Fig 1 to explain some of the results.*

I list some of these problems below.

(Comment 73) Beyond the main figures, the ordering of the Supplementary Figures and Notes does not match the point in which these materials are referenced in the main text, which unnecessarily complicates things.

Thanks for this. We have revised the ordering of supplementary information accordingly, and have also removed duplicate content in the SI. In one case, we elected to retain an out-of-order reference in the revised SI - Supplementary Notes 3 logically follows 1-2 from a methodological perspective, but is referenced in the Results and Discussion section of the main text. We retain this arrangement because it is in our view the most logical presentation.

(Comment 74) Fig1E: it is not clear what the vertical blue/black lines mean in this context.

We have more clearly annotated the figure to clarify this issue. We aimed to illustrate for each non alphaproteobacterial gene family its divergence date from other bacteria (black bar) and its associated pre-LECA duplications (blue bars). The horizontal line serves to connect these data points, and we hope the revised figure (and its additional text) clarifies this intent.

(Comment 75) Fig2. The meaning of the green dots, squares and diamonds in the figure should be explained in the legend? What does “HDP” mean in this context? (presumably something related to the density distribution, but this should be made clear). There are no gene families indicated with “ α ” — do the authors mean “a” or “A”, as this refers to asgard archaea? There are similar issues in Fig3.

(i) Gene family origins- we have decided to include a key on the figures, as well as making the symbols a bit more distinct. To help clarify these points:

- *Differently coloured bacterial and archaeal gene families*
- *Show directly in the key what these nodes look like*

(ii) The key now also describes the green shapes “similarly sided shapes connect duplication events suspected to have a similar age”

(iii) We have corrected the typo, but rather more generally HPD, and CI were used inconsistently in the manuscript, we have corrected this and now consistently use CI (Confidence Interval) throughout the MS.

Comments on the supplementary materials/notes

(Comment 76) In Supplementary Notes 1 and 2 the methods are explained with an adequate level of detail, but in a very schematic way that complicates things unnecessarily. Some introductory remarks describing what is to be found in each section would be welcome. I acknowledge that this is provided in the main text summarised Methods section, but going back and forth between the two documents is an unnecessary hassle.

We have now extensively revised the SI along the lines suggested, including an introduction at the beginning of each section. We have also moved text specific to the running of scripts to readme files that can be found in the same directory as the respective scripts.

(Comment 77) In Supplementary Note 3: “determine the impact of alternatives on our results.” Be more explicit: alternative roots?

We have made some changes to the layout to Supplementary Notes 3, each section now has a prefacing sentence describing the tests and what they were set out to explore. Specifically with regard to this point its section is now titled: Sensitivity of inferred eukaryogenesis timescale to alternative tree topologies and calibrations

“In the first set of tests we explored a number of tree topological and calibration alternatives which might affect our eukaryogenesis timeline, specifically the effect of using an unconstrained bacterial root, over the asserted root in our species tree, the effect of the total-group Oxyphotobacteria calibration on the age of LBCA, and the effect of current competing eukaryotic root topologies (Metamonad1, Opimoda / Diphoda1,2) on the age of LECA. These tests were performed by producing modified input files for MCMCTree and time-resolving them under the same conditions as the

species tree. These results are tabulated in Supplementary Table 4, in summary our findings indicated:”
Supplementary Notes 3

(Comment 78) Sadly, some figures in the supplementary notes are not provided with sufficient resolution (e.g. species trees such as Supplementary Figure 7-9). This should be revised.

Agreed. As these trees contain hundreds of tips, we have decided to instead provide them as Newick-formatted tree files and larger PDF renderings in the Supplementary Data repository rather than directly in the SI.

Referee #5 (Remarks on code availability):

(Comment 79) The supplementary notes contain some inline code invocations, but they also reference some scripts and directory structures which I could not find in the SM or any associated data repository.

Thanks we have revised the SM accordingly, more generally we have tried to improve the structure of the SI by:

- (i) Simplifying the directory structure, such that what each folder is about is clear.*
- (ii) In most directories including a readme text which described the directory contents.*
- (iii) Reducing the number of files per directory, and archiving directories which have a lot of files (domain identification / analysis reports, MCMCTree runs and visualisations), which should make finding scripts easier, as well as making the supplementary information a bit more convenient to move around.*

Response to remaining issues:

1. Please submit a revised title within 75 characters (including spaces) that is free of any punctuation marks such as colons, exclamation marks, full stops or speech marks.

We have changed the title from

Dated gene duplications elucidate the evolutionary assembly of the eukaryotic cell (82 chars)

to

Dated gene duplications elucidate the evolutionary assembly of eukaryotes (72 chars)

2. Please add references to the abstract.

We have added 9 references to the abstract.

3. The number of main text references should be 60 in total or less - currently there are 117.

We have reduced the number of references to 68. This is the bare minimum. Further reduction will compromise attribution of IP and leave us with ex cathedra statements.

4. Flagging that there are no methods references - please create a separate reference list for the methods with continuous numbering (this would be a good place to put your excess references from the main-text reference section).

A methods reference section has been added.

5. Please remove the main figures from the article file and re-supply them individually in an acceptable format such as EPS, AI, PS, PDF, PPT, PSD or XLS (for graphs) with editable vector files.

We have removed the figures from the manuscript and provide PDF versions of them (in Manuscript_figures.pdf)

6. Flagging that there are potential third party rights issues in the figures - please check sources or if permissions are needed for the cells and schematic of cell signalling illustrations in the figures.

All figures were designed by the author Christopher Kay, no figures contain material requiring a third party rights declaration.

8. Please provide a data availability statement in the main text of the manuscript.

We have added a data availability statement:

Data Availability

The dataset generated in the course of this study are available at the University of Bristol data repository, [data.bris](https://data.bris.ac.uk/), at <https://doi.org/10.5523/bris.tjfgfs0kmr532t2gvqpcdbmto>.

9. Please ensure that the text size in all figures is at least 5 pt Arial.

We verify all text within the figures is at least size 5pt

10. For any Supplementary Figures, please check and confirm that:

- * If data is presented as bar charts, individual data points are shown using overlaid dot plots.
- * The n number (i.e. the sample size used to derive statistics) is provided and defined as a precise value (not a range), using the wording “n=X samples/cells/independent experiments” etc. where applicable.
- * Any chart axis, error bars, scale bars, symbols and colour scales are defined.
- * Any statistical tests used for data analysis are specified and exact p-values are provided either on the figures themselves, in the legend or in the Source Data file.
- * Wherever representative data such as micrographs are shown, the legend indicates how many times the experiment was repeated with the same results.

We confirm that our supplementary figures conform to these specifications.

11. Please note that a statement on code availability is missing from the manuscript. Further, there is a reference of custom scripts in methods section of the manuscript. Further, in the ‘Software and codes’ section of the RS authors have mentioned that, ‘Descriptions of how to run the programs, their requirements, sample software environments and sample data are provided in the Supplementary Material, which is stored on the public archive Zenodo.’ Please provide the accessible Zenodo weblink under separate ‘Code availability’ section of the manuscript.

We have added a data availability statement:

Code Availability

The code used to generate our results is provided with the data stored at the University of Bristol’s data.bris research data repository with the identifier doi: <https://doi.org/10.5523/bris.tjfgfs0kmr532t2gvqpcdbmto>.